# SmoQyDQMC.jl: A flexible implementation of determinant quantum Monte Carlo for Hubbard and electron-phonon interactions

Benjamin Cohen-Stead[1,2], Sohan Malkaruge Costa[1,2], James Neuhaus[1,2],
Andy Tanjaroon Ly[1,2], Yutan Zhang[3], Richard Scalettar[3], Kipton Barros[4], and
Steven Johnston[1,2,⋆]

**1** Department of Physics and Astronomy, The University of Tennessee, Knoxville, Tennessee 37996, USA

**2** Institute for Advanced Materials and Manufacturing, The University of Tennessee, Knoxville, Tennessee 37996, USA

**3** Department of Physics and Astronomy, University of California, Davis, California 95616, USA

**4** Theoretical Division and CNLS, Los Alamos National Laboratory, Los Alamos, New Mexico 87545, USA

⋆ sjohn145@utk.edu

April 18, 2024

## Abstract

We introduce the `SmoQyDQMC.jl` package, a Julia implementation of the determinant quantum Monte Carlo algorithm. `SmoQyDQMC.jl` supports generalized tight-binding Hamiltonians with on-site Hubbard and generalized electron-phonon ($e$-ph) interactions, including non-linear $e$-ph coupling and anharmonic lattice potentials. Our implementation uses hybrid Monte Carlo methods with exact forces for sampling the phonon fields, enabling efficient simulation of low-energy phonon branches, including acoustic phonons. The `SmoQyDQMC.jl` package also uses a flexible scripting interface, allowing users to adapt it to different workflows and interface with other software packages in the Julia ecosystem. The code for this package can be downloaded from our GitHub repository at https://github.com/SmoQySuite/SmoQyDQMC.jl or installed using the Julia package manager. The online documentation, including examples, can be obtained from our document page at https://smoqysuite.github.io/SmoQyDQMC.jl/stable/.

# 1 Introduction

## 1.1 Overview & Scope

This paper introduces `SmoQyDQMC.jl`, a user-friendly Julia implementation of the determinant quantum Monte Carlo (DQMC) algorithm [1,2], and its associated auxiliary packages.[1] The `SmoQyDQMC.jl` package is capable of simulating a broad class of tight-binding Hamiltonians with on-site Hubbard and/or generalized electron-phonon ($e$-ph) interactions. The model Hamiltonians can be defined on arbitrary lattice geometries and a wide variety of measurements can be performed for these simulations, including density-density, spin-spin, current-current correlation functions and pairing for different symmetries. The code has been designed with a scripting interface inspired by packages like `ITensor.jl` [3], rather than a configuration file-based workflow commonly found in open-source implementations of DQMC [4,5]. This design allows users to easily incorporate `SmoQyDQMC.jl` into complex workflows and directly interface it with the Julia programming language's rich ecosystem of scientific computing and machine learning packages. Crucially, our implementation accomplishes this flexibility without sacrificing performance, with `SmoQyDQMC.jl` achieving ideal $O(\beta N^3)$ scaling in computational complexity in the system size $N$ and inverse temperature $\beta$.

    This document provides detailed descriptions of `SmoQyDQMC.jl`'s design philosophy and

---

[1]The names for both the `SmoQyDQMC.jl` package, and the overarching SmoQy Suite GitHub organization it lives in, are inspired by the Great Smoky Mountain range running along the Tennessee–North Carolina border in the southeastern United States.

underlying algorithms. It is also intended to serve as a cite-able document when using this code for research. This document is not intended to serve as a detailed user manual for the package. Instead, we encourage readers to consult our online documentation, which will be maintained as a living document with new examples and reference material added continuously over the package's lifetime.

## 1.2 Background and Motivation

Quantum Monte Carlo (QMC) algorithms are a powerful family of methods for performing numerically exact nonperturbative simulations of quantum many-body condensed matter systems [6–35]. These methods come in many flavors, including zero temperature projection and variational Monte Carlo methods, to finite-temperature auxiliary field methods like DQMC or continuous-time QMC and beyond.

DQMC is an auxiliary-field QMC method [1, 2], which calculates expectation values of a quantum system within the grand canonical ensemble. The method has been applied to a broad class of problems in condensed matter physics, including single- [2, 36–44] and multi-band Hubbard models [45–51], negative-$U$ models [52–55], $e$-ph coupled Hamiltonians like the Holstein [56–60] and Su-Schrieffer-Heeger (SSH) models [61–66] and their strongly correlated counterparts [67–70], and topological systems [71–76]. It has also been used to simulate ultra-cold atom experiments [77–82], quantum entanglement [83–85] and beyond [86–89].

Several open-source implementations of the DQMC algorithm have been developed over the years with support for different classes of Hamiltonians. Perhaps the most popular and well-known are the *Algorithms for Lattice Fermions* (ALF) [5] and *QUantum Electron Simulation Toolbox* (QUEST) [4] projects. The QUEST package supports single- and multi-orbital Hubbard Hamiltonians defined on arbitrary lattice geometries. The ALF package provides support for a much broader class of Fermionic interactions, as well as coupling to classical or quantum bosonic fields, which enables simulations of standard $e$-ph Hamiltonians like the Holstein model. Both QUEST and ALF are currently implemented in Fortran90, which hinders their integration with modern machine learning and scientific computing packages.[2]

The `SmoQyDQMC.jl` package is a Julia implementation of the DQMC algorithm, with support for tight-binding Hubbard Hamiltonians with and without $e$-ph interactions. The code leverages hybrid Monte Carlo (HMC) methods [91–93] to sample the phonon fields, which allows it to treat a much broader class of $e$-ph interactions. The package also includes additional features that are not currently included in existing DQMC implementations. Specifically, the initial release of this package includes support for

1. arbitrary lattices and bases in zero-, one-, two-, and three-dimensions;
2. intra-orbital Hubbard interactions in any site/orbital in the lattice;
3. real or complex hopping parameters to enable the introduction of magnetic fields or boundary condition averaging to reduce finite size effects;
4. Hamiltonians with spin-dependent tight-binding and $e$-ph interaction parameters;
5. fully momentum-dependent $e$-ph interactions, including long-range Holstein- and SSH-like couplings;
6. coupling to multiple phonon branches, either via the same or different microscopic coupling mechanisms;
7. low-energy optical and acoustic phonon branches;
8. nonlinear $e$-ph interactions and anharmonic lattice potentials up to fourth order in the atomic displacements;

---

[2]PyALF [90], a high-level python wrapper for ALF, is currently in development but has not been formally published at this time.

9. dynamical tuning of the chemical potential to archive a targeted density $\langle n \rangle$;

10. spatial disorder in any Hamiltonian parameter; and

11. support for measuring a wide range of common observables on arbitrary lattice geometries.

Importantly, `SmoQyDQMC.jl` is user friendly and utilizes a scripting interface rather than more traditional workflows based on input configuration files. This design allows users to implement straightforward parallelization at the script level, adapt the package to their existing workflows, and more readily interface `SmoQyDQMC.jl` with the rich ecosystem of scientific computing packages being actively developed in the Julia programming language. For example, it can be readily coupled to existing machine learning and artificial intelligence packages to enable new research in this direction [94].

If the `SmoQyDQMC.jl` package is unable to meet the needs of a particular user, we have also provided two lower-level supporting packages. The first package, `JDQMCFrameworks.jl`, implements the core computational kernel of the DQMC algorithm. The second package, `JDQMCMeasurements.jl`, implements a set of functions for measuring various correlation functions for arbitrary lattice geometries in a DQMC simulation. This package also exports several additional utility functions for transforming measurements from position space to momentum space, and also measuring susceptibilities by integrating correlation functions over the imaginary time axis. These two lower-level packages were used to develop the `SmoQyDQMC.jl` package, and provide the tools necessary for a user to develop their own specialized implementations of the DQMC algorithm.

## 1.3 Relevant Links, Documentation, and Reporting

The source code for the `SmoQyDQMC.jl` package and its associated auxiliary packages can be found on the SmoQy Suite's GitHub page [95]. As a package registered with Julia programming language's General registry, the package can also be installed using the Julia package manager by issuing the command

```
julia> ]
pkg> add SmoQyDQMC
```

The package's documentation, including examples, can be found in our online documenation [96]. It includes full API and a growing list of example scripts that can be used to run several model Hamiltonians of interest. The source code for the `JDQMCFrameworks.jl` [97] and `JDQMCMeasurements.jl` [98] packages can also be found on the SmoQy Suite's GitHub page.

## 2 Supported Hamiltonians

### 2.1 Specifications

This section describes the class of Hamiltonians currently supported by the `SmoQyDQMC.jl` package, and how the various terms appearing in the Hamiltonian are parameterized within the code base. Throughout this section, we use bold roman indices (e.g., $\mathbf{i}$, $\mathbf{j}$, ...) to index the unit cell within the cluster, Greek symbols (e.g., $\nu$, $\gamma$, $\mu$, ...) to index different orbitals within the atomic basis, and $n_{\mathbf{i},\nu}$ to index different phonon species on orbital $\nu$ of unit cell $\mathbf{i}$. Throughout, we normalize $\hbar = 1$ and denote $\mathcal{N} = N \cdot n$ as the total number of orbitals in the lattice, where $N$ is the number of unit cells, and $n$ is the number of orbitals per unit cell.

For our purposes, it is convenient to partition the full Hamiltonian into three terms

$$\hat{\mathcal{H}} = \hat{\mathcal{U}} + \hat{\mathcal{K}} + \hat{\mathcal{V}}, \tag{1}$$

where $\hat{\mathcal{U}}$ describes the non-interacting lattice (phonon) degrees of freedom and $\hat{\mathcal{K}}$ and $\hat{\mathcal{V}}$ are the total electron kinetic and potential energies, respectively. Note that both $\hat{\mathcal{K}}$ and $\hat{\mathcal{V}}$ can depend on the dynamical lattice coordinates, leading to an electron-phonon coupling that is either diagonal or off-diagonal in the orbital basis. $\hat{\mathcal{V}}$ also includes any contributions from the intra-orbital Hubbard repulsion on a given site.

The non-interacting lattice terms are further subdivided into the sum of three terms

$$\hat{\mathcal{U}} = \hat{\mathcal{U}}_{\text{qho}} + \hat{\mathcal{U}}_{\text{anh}} + \hat{\mathcal{U}}_{\text{disp}}. \tag{2}$$

The first term

$$\hat{\mathcal{U}}_{\text{qho}} = \sum_{\mathbf{i},\nu} \sum_{n_{\mathbf{i},\nu}} \left[ \frac{1}{2M_{n_{\mathbf{i},\nu}}} \hat{P}^2_{n_{\mathbf{i},\nu}} + \frac{1}{2} M_{n_{\mathbf{i},\nu}} \Omega^2_{0,n_{\mathbf{i},\nu}} \hat{X}^2_{n_{\mathbf{i},\nu}} \right] \tag{3}$$

describes the placement of local quantum harmonic oscillator (QHO) modes on sites in the lattice, i.e. an Einstein solid, while the second term

$$\hat{\mathcal{U}}_{\text{anh}} = \sum_{\mathbf{i},\nu} \sum_{n_{\mathbf{i},\nu}} \left[ \frac{1}{24} M_{n_{\mathbf{i},\nu}} \Omega^2_{a,n_{\mathbf{i},\nu}} \hat{X}^4_{n_{\mathbf{i},\nu}} \right] \tag{4}$$

introduces anharmonic contributions to the oscillator potential. The third term

$$\hat{\mathcal{U}}_{\text{disp}} = \sum_{\substack{\mathbf{i},\nu \\ \mathbf{j},\gamma}} \sum_{\substack{n_{\mathbf{i},\nu} \\ n_{\mathbf{j},\gamma}}} \frac{M_{n_{\mathbf{i},\alpha}} M_{n_{\mathbf{j},\gamma}}}{M_{n_{\mathbf{i},\alpha}} + M_{n_{\mathbf{j},\gamma}}} \left[ \tilde{\Omega}^2_{0,n_{\mathbf{i},\alpha},n_{\mathbf{j},\gamma}} (\hat{X}_{n_{\mathbf{i},\nu}} - \hat{X}_{n_{\mathbf{j},\gamma}})^2 + \frac{1}{12} \tilde{\Omega}^2_{a,n_{\mathbf{i},\alpha},n_{\mathbf{j},\gamma}} (\hat{X}_{n_{\mathbf{i},\nu}} - \hat{X}_{n_{\mathbf{j},\gamma}})^4 \right] \tag{5}$$

introduces coupling (or dispersion) between the QHO modes. The sums over $\mathbf{i}$ ($\mathbf{j}$) and $\nu$ ($\gamma$) run over unit cells in the lattice and basis orbitals within each unit cell, respectively. The sum over $n_{\mathbf{i},\nu}$ ($n_{\mathbf{j},\gamma}$) runs over the different flavors of QHO modes placed on a given orbital in the lattice.

The position and momentum operators for each QHO mode are $\hat{X}_{n_{\mathbf{i},\nu}}$ and $\hat{P}_{n_{\mathbf{i},\nu}}$ respectively, with corresponding phonon mass $M_{n_{\mathbf{i},\nu}}$. The spring constant is $K_{n_{\mathbf{i},\nu}} = M_{n_{\mathbf{i},\nu}} \Omega^2_{0,n_{\mathbf{i},\nu}}$, with $\Omega_{0,n_{\mathbf{i},\nu}}$ specifying the phonon frequency. $\hat{\mathcal{U}}_{\text{anh}}$ then introduces an anharmonic $\hat{X}^4_{n_{\mathbf{i},\nu}}$ contribution to the QHO potential energy that is controlled by the parameter $\Omega_{a,n_{\mathbf{i},\nu}}$. Similarly, $\tilde{\Omega}_{0,n_{\mathbf{i},\alpha},n_{\mathbf{j},\gamma}}$ ($\tilde{\Omega}_{a,n_{\mathbf{i},\alpha},n_{\mathbf{j},\gamma}}$) is the coefficient controlling harmonic (anharmonic) dispersion between QHO modes.

The electron kinetic energy is conveniently expressed as

$$\hat{\mathcal{K}} = \hat{\mathcal{K}}_0 + \hat{\mathcal{K}}_{\text{ssh}} = \sum_\sigma \hat{\mathcal{K}}_{0,\sigma} + \sum_\sigma \hat{\mathcal{K}}_{\text{ssh},\sigma}. \tag{6}$$

The first term describes the non-interacting spin-$\sigma$ electron kinetic energy

$$\hat{\mathcal{K}}_{0,\sigma} = -\sum_{\substack{\mathbf{i},\nu \\ \mathbf{j},\gamma}} \left[ t_{\sigma,(\mathbf{i},\nu),(\mathbf{j},\gamma)} \hat{c}^\dagger_{\sigma,\mathbf{i},\nu} \hat{c}_{\sigma,\mathbf{j},\gamma} + \text{h.c.} \right], \tag{7}$$

where $t_{\sigma,(\mathbf{i},\nu),(\mathbf{j},\gamma)}$ is the spin-$\sigma$ hopping integral from orbital $\gamma$ in unit cell $\mathbf{j}$ to orbital $\nu$ in unit cell $\mathbf{i}$, and may be real or complex. The second term describes the interaction between the

lattice degrees of freedom and the spin-$\sigma$ electron kinetic energy via a Su-Schrieffer-Heeger (SSH)-like coupling mechanism [99, 100]

$$\hat{\mathcal{K}}_{\text{ssh},\sigma} = \sum_{\substack{\mathbf{i},\nu \\ \mathbf{j},\gamma}} \sum_{\substack{n_{\mathbf{i},\nu} \\ n_{\mathbf{j},\gamma}}} \sum_{m=1}^{4} (\hat{X}_{n_{\mathbf{i},\nu}} - \hat{X}_{n_{\mathbf{j},\gamma}})^m \left[ \alpha_{\sigma,m,n_{\mathbf{i},\nu},n_{\mathbf{j},\gamma}} \hat{c}^{\dagger}_{\sigma,\mathbf{i},\nu} \hat{c}_{\sigma,\mathbf{j},\gamma} + \text{h.c.} \right]. \tag{8}$$

Here, the modulations of the spin-$\sigma$ hopping integrals to $m^{\text{th}}$ ($= 1-4$) order in displacement are controlled by the parameters $\alpha_{\sigma,m,n_{\mathbf{i},\nu},n_{\mathbf{j},\gamma}}$.

Lastly, the electron potential energy is expressed as

$$\hat{\mathcal{V}} = \hat{\mathcal{V}}_0 + \hat{\mathcal{V}}_{\text{hol}} + \hat{\mathcal{V}}_{\text{hub}} = \sum_{\sigma} \hat{\mathcal{V}}_{0,\sigma} + \sum_{\sigma} \hat{\mathcal{V}}_{\text{hol},\sigma} + \hat{\mathcal{V}}_{\text{hub}}, \tag{9}$$

where

$$\hat{\mathcal{V}}_{0,\sigma} = \sum_{\mathbf{i},\nu} \left[ (\epsilon_{\sigma,\mathbf{i},\nu} - \mu) \hat{n}_{\sigma,\mathbf{i},\nu} \right] \tag{10}$$

is the non-interacting spin-$\sigma$ electron potential energy. Here, $\mu$ is the chemical potential and $\epsilon_{\sigma,\mathbf{i},\nu}$ is the spin-$\sigma$ on-site energy for orbital $\nu$ in unit cell $\mathbf{i}$.

The second term

$$\hat{\mathcal{V}}_{\sigma,\text{hol}} = \begin{cases} \sum_{\substack{\mathbf{i},\nu \\ \mathbf{j},\gamma}} \sum_{n_{\mathbf{i},\nu}} \left[ \sum_{m=1,3} \tilde{\alpha}_{\sigma,m,n_{\mathbf{i},\nu},(\mathbf{j},\gamma)} \hat{X}^m_{n_{\mathbf{i},\nu}} (\hat{n}_{\sigma,\mathbf{j},\gamma} - \frac{1}{2}) + \sum_{m=2,4} \tilde{\alpha}_{\sigma,m,n_{\mathbf{i},\nu},(\mathbf{j},\gamma)} \hat{X}^m_{n_{\mathbf{i},\nu}} \hat{n}_{\sigma,\mathbf{j},\gamma} \right] \\ \sum_{\substack{\mathbf{i},\nu \\ \mathbf{j},\gamma}} \sum_{n_{\mathbf{i},\nu}} \sum_{m=1}^{4} \tilde{\alpha}_{\sigma,m,n_{\mathbf{i},\nu},(\mathbf{j},\gamma)} \hat{X}^m_{n_{\mathbf{i},\nu}} \hat{n}_{\sigma,\mathbf{j},\gamma} \end{cases} \tag{11}$$

is the contribution to the spin-$\sigma$ electron potential energy that results from a Holstein- or Fröhlich-like coupling to the lattice degrees of freedom. The parameter $\tilde{\alpha}_{\sigma,m,n_{\mathbf{i},\nu},(\mathbf{j},\gamma)}$ controls the strength this coupling in the $\hat{\mathcal{V}}_{\sigma,\text{hol}}$ term. It is important to note that the two available parametrizations shown in Eq. (11) that are available in `SmoQyDQMC.jl` are inequivalent.

Finally, the third term

$$\hat{\mathcal{V}}_{\text{hub}} = \begin{cases} \sum_{\mathbf{i},\nu} U_{\mathbf{i},\nu} \left( \hat{n}_{\uparrow,\mathbf{i},\nu} - \frac{1}{2} \right) \left( \hat{n}_{\downarrow,\mathbf{i},\nu} - \frac{1}{2} \right) \\ \sum_{\mathbf{i},\nu} U_{\mathbf{i},\nu} \hat{n}_{\uparrow,\mathbf{i},\nu} \hat{n}_{\downarrow,\mathbf{i},\nu} \end{cases} \tag{12}$$

is the on-site Hubbard interaction contribution, where $U_{\mathbf{i},\nu}$ is the on-site Hubbard interaction strength. Note that `SmoQyDQMC.jl` allows the user to parameterize the Hubbard interaction using either functional form for $\hat{\mathcal{V}}_{\text{hub}}$. The top-most is particle-hole symmetric and is often useful at half-filling.

## 2.2 A Flexible Approach to Electron-Phonon Coupling

The class of Hamiltonians supported by `SmoQyDQMC.jl` is flexible enough to accommodate the simulation of most standard $e$-ph model Hamiltonians. For example, the canonical single-band Holstein model with coupling to a single Einstein phonon branch can be obtained by placing a single QHO mode on each site ($n_{\mathbf{i},\nu} = 1$) and setting $\hat{\mathcal{U}}_{\text{anh}} = \hat{\mathcal{U}}_{\text{disp}} = \hat{\mathcal{K}}_{\text{ssh}} = 0$, while retaining the $\hat{\mathcal{V}}_{\text{hol}}$ term [101]. Similarly, the optical single-band SSH model in $D$-dimensions [64] can be arrived at by setting $\hat{\mathcal{U}}_{\text{anh}} = \hat{\mathcal{U}}_{\text{disp}} = \hat{\mathcal{V}}_{\text{hol}} = 0$ but requires $D$ QHO modes per site. The bond SSH model can also be expressed by coupling pairs of QHO modes, one with finite mass and the other with infinite mass, effectively associating the finite mass QHO mode with a bond [64, 102]. The acoustic SSH model can be expressed by introducing QHO modes with zero frequency ($\Omega_{0,n_{\mathbf{i},\nu}} = 0$), that are then coupled together with the $\hat{\mathcal{U}}_{\text{disp}}$ term [100]. Importantly, `SmoQyDQMC.jl` allows users to define Hamiltonians that combine these various models, enabling the simulation of systems with multiple phonon branches that can each couple to the electrons in different ways.

# 3  Algorithm Details

This section provides details on the various algorithms used in the `SmoQyDQMC.jl` package. Here, we have taken an axiomatic approach and focused on describing what the algorithms do rather than providing detailed derivations or justifications for their correctness. Instead, we have provided several references throughout the text for any reader who is interested in the relevant derivations.

## 3.1  Formulation of DQMC Algorithm

DQMC is an auxiliary field QMC method for simulating systems of itinerant fermions on a lattice in the grand canonical ensemble [1,2]. For a inverse temperature $\beta = 1/T$ ($k_{\mathrm{B}} = 1$), the algorithm adopts a discrete imaginary time grid $\tau = \Delta\tau \cdot l$, where $l = 1, \dots, L_\tau$ indexes the imaginary time-slice and $\Delta\tau = \beta/L_\tau$. Using this grid, DQMC then expresses the partition function as a path-integral in imaginary time

$$\mathcal{Z} = \mathrm{Tr}\left[\prod_{l=1}^{L_\tau} \hat{\mathcal{B}}\right], \tag{13}$$

where $\hat{\mathcal{B}} = e^{-\Delta\tau\hat{\mathcal{H}}}$ is the imaginary-time propagator over the discretization interval $\Delta\tau$.

Next, the Suzuki-Trotter (ST) approximation is applied to all imaginary-time slices $l$ to factorize the various terms appearing in the exponentiated Hamiltonian [103]. The resulting form for the partition function is

$$\mathcal{Z} \approx \mathrm{Tr}\left[\prod_{l=1}^{L_\tau} \hat{B}\right] + \mathcal{O}(\Delta\tau^2), \tag{14}$$

where the $\hat{\mathcal{B}} \approx \hat{B} + \mathcal{O}(\Delta\tau^3)$ approximation

$$\hat{B} = e^{-\frac{\Delta\tau}{2}\hat{\mathcal{K}}} e^{-\Delta\tau\hat{\mathcal{V}}} e^{-\frac{\Delta\tau}{2}\hat{\mathcal{K}}} \tag{15}$$

is used. By applying cyclic property of the trace it is straightforward to see that Eq. (14) is left unchanged if the lower order $\hat{\mathcal{B}} \approx \hat{B} + \mathcal{O}(\Delta\tau^2)$ approximation

$$\hat{B} = e^{-\Delta\tau\hat{\mathcal{V}}} e^{-\Delta\tau\hat{\mathcal{K}}} \tag{16}$$

is used instead. `SmoQyDQMC.jl` can run simulations based on applying either Eq. (15) or (16). In cases where the kinetic energy operator $\hat{\mathcal{K}}$ is static, a formulation based on Eq. (16) is usually preferable as the computational overhead is lower. However, when $e$-ph interactions modulate the hopping amplitudes and the checkerboard approximation is applied, it is better to use a formulation based on Eq. (15) as it improves the efficacy of this additional approximation; this aspect is discussed in greater detail in Sec. 3.5.

At this point, the only term in the Hamiltonian that is not quadratic in Fermion creation and annihilation operators is $\hat{\mathcal{V}}_{\mathrm{hub}}$. This term can be rendered quadratic using the discrete Ising Hubbard-Stratonovich (HS) transformation [104,105] (or other variants [106–108])

$$e^{-\frac{1}{4}\Delta\tau U \hat{m}_\uparrow \hat{m}_\downarrow} = \frac{1}{2} e^{-\frac{1}{4}\Delta\tau|U|} \sum_{s=\pm 1} e^{\frac{\alpha s}{2}(\hat{m}_\uparrow - \eta\hat{m}_\downarrow)}, \tag{17}$$

where $\hat{m}_\sigma = 2\hat{n}_\sigma - 1$, $\eta = U/|U|$, and $\cosh\alpha = e^{\frac{1}{2}\Delta\tau|U|}$. This decoupling is valid for both a repulsive ($U > 0$) and attractive ($U < 0$) local Hubbard interaction. Applying the Ising HS

transformation at each imaginary-time introduces Ising HS variables $s_{l,\mathbf{i},\nu}$ at every orbital with a finite Hubbard interaction in the lattice, for every imaginary time slice $l$.

Next, we evaluate the trace over the lattice degrees of freedom. This task is best accomplished in the position basis where we can analytically integrate out the phonon momentum operators $\hat{P}_{n_{\mathbf{i},\nu}}$ and replace the phonon position operators $\hat{X}_{n_{\mathbf{i},\nu}}$ by scalar phonon fields $x_{l,n_{\mathbf{i},\nu}}$ for all phonon modes $n_{\mathbf{i},\nu}$ and imaginary-time slices $l$ [56, 61, 63, 67].

After these operations are performed, each term appearing in the exponentiated Hamiltonian is quadratic in Fermionic operators, and may be written in the form

$$\hat{O}_{\sigma,l} = \hat{\mathbf{c}}_\sigma^\dagger O_{\sigma,l} \hat{\mathbf{c}}_\sigma, \tag{18}$$

where $\hat{\mathbf{c}}_\sigma^\dagger = [\hat{c}_{\sigma,1,1}^\dagger, \ldots, \hat{c}_{\sigma,N,n}^\dagger]$ is a row-vector of creation operators for each of the $\mathcal{N}$ orbitals in the system, and $\hat{\mathbf{c}}_\sigma$ is the corresponding column vector of annihilation operators. Therefore, $O_{\sigma,l}$ is a $\mathcal{N} \times \mathcal{N}$ Hermitian matrix.

The resulting expression for the partition function is

$$\mathcal{Z} \approx \sum_s \int \mathcal{D}x \prod_{\sigma=\uparrow,\downarrow} \prod_{l=1}^{L_\tau} \hat{B}_{\sigma,l}, \tag{19}$$

with $\sum_s$ and $\int \mathcal{D}x$ denoting the path integral over all Ising HS fields $s_{l,\mathbf{i},\nu}$ and all phonon fields $x_{l,n_{\mathbf{i},\nu}}$, respectively. The propagator operators now explicitly depend on the spin $\sigma$ and imaginary-time slice $l$ and are given by

$$\hat{B}_{\sigma,l} = e^{-\frac{\Delta\tau}{2}\hat{\mathcal{K}}_{\sigma,l}} e^{-\Delta\tau\hat{\mathcal{V}}_{\sigma,l}} e^{-\frac{\Delta\tau}{2}\hat{\mathcal{K}}_{\sigma,l}}, \tag{20}$$

or

$$\hat{B}_{\sigma,l} = e^{-\Delta\tau\hat{\mathcal{V}}_{\sigma,l}} e^{-\Delta\tau\hat{\mathcal{K}}_{\sigma,l}}, \tag{21}$$

depending on whether the approximation from Eq. (15) or Eq. (16) is used, respectively. The dependence on the fields $s_{l,\mathbf{i},\nu}$ and $x_{l,n_{\mathbf{i},\nu}}$ appear as

$$\hat{\mathcal{K}}_{\sigma,l} = \left[\hat{\mathcal{K}}_{0,\sigma} + \hat{\mathcal{K}}_{\text{ssh},\sigma,l}\right] = \left[\hat{\mathcal{K}}_{0,\sigma} + \hat{\mathcal{K}}_{\text{ssh},\sigma}(x_{l,n_{\mathbf{i},\nu}})\right] \tag{22}$$

in the electron kinetic energy, and

$$\hat{\mathcal{V}}_{\sigma,l} = \left[\hat{\mathcal{V}}_{0,\sigma} + \hat{\mathcal{V}}_{\text{hol},\sigma,l} + \hat{\mathcal{V}}_{\text{hub},\sigma,l}\right] = \left[\hat{\mathcal{V}}_{0,\sigma} + \hat{\mathcal{V}}_{\text{hol},\sigma}(x_{l,n_{\mathbf{i},\nu}}) + \hat{\mathcal{V}}_{\text{hub},\sigma}(s_{l,\mathbf{i},\nu})\right] \tag{23}$$

in the electron potential energy, respectively.

Employing the Blankenbecler, Scalapino and Sugar (BSS) relation [1], we integrate out the Fermionic degrees of freedom to arrive at our final expression for the partition function

$$\mathcal{Z} \approx \sum_s \int \mathcal{D}x \; e^{-S_{\text{B}}(x)} \prod_{\sigma=\uparrow,\downarrow} \det M_\sigma(\tau) \; + \; \mathcal{O}(\Delta\tau^2), \tag{24}$$

where $S_{\text{B}}(x)$ is the strictly bosonic action associated with the lattice degrees of freedom and $M_\sigma(\tau)$ is the spin-$\sigma$ Fermion determinant matrix. The latter is given by

$$M_\sigma(\tau) = I + B_{\sigma,l} B_{\sigma,l-1} \ldots B_{\sigma,1} B_{\sigma,L_\tau} \ldots B_{\sigma,l+1}, \tag{25}$$

where $I$ is the identity matrix, and $B_{\sigma,l}$ are the spin-$\sigma$ propagator matrices for imaginary-time slice $l$. Note that $\det M_\sigma(\tau)$ is the same for all values of $\tau$. The propagator matrices take a similar form to the propagator operators appearing in Eq. (20) and Eq. (21), with

$$B_{\sigma,l} = e^{-\frac{\Delta\tau}{2}K_{\sigma,l}} e^{-\Delta\tau V_{\sigma,l}} e^{-\frac{\Delta\tau}{2}K_{\sigma,l}}, \tag{26}$$

or

$$B_{\sigma,l} = e^{-\Delta\tau V_{\sigma,l}} e^{-\Delta\tau K_{\sigma,l}}, \tag{27}$$

where $K_{\sigma,l}$ and $V_{\sigma,l}$ are the spin-$\sigma$ electron kinetic and potential energy matrices for imaginary-time slice $l$, respectively. Each $K_{\sigma,l}$ is strictly off-diagonal and Hermitian, while the $V_{\sigma,l}$ matrices are diagonal.

The total bosonic action $S_B(x)$ appearing in Eq. (24) can be conveniently expressed as a sum of four terms

$$S_B(x) = S_{qho}(x) + S_{anh}(x) + S_{disp}(x) + S_{hol}(x). \tag{28}$$

The first term

$$S_{qho}(x) = \Delta\tau \sum_{\mathbf{i},\nu} \sum_{n_{\mathbf{i},\nu}} \sum_{l} \left[ \frac{M_{n_{\mathbf{i},\nu}} \Omega^2_{0,n_{\mathbf{i},\nu}}}{2} x^2_{l,n_{\mathbf{i},\nu}} + \frac{M_{n_{\mathbf{i},\nu}}}{2} \left( \frac{x_{l+1,n_{\mathbf{i},\nu}} - x_{l,n_{\mathbf{i},\nu}}}{\Delta\tau} \right)^2 \right] \tag{29}$$

is the contribution to the bosonic action arising from the QHO term $\hat{\mathcal{U}}_{qho}$ in the Hamiltonian, as defined by Eq. (3), and the term

$$S_{anh}(x) = \Delta\tau \sum_{\mathbf{i},\nu} \sum_{n_{\mathbf{i},\nu}} \sum_{l} \frac{M_{n_{\mathbf{i},\nu}} \Omega^2_{a,n_{\mathbf{i},\nu}}}{24} x^4_{l,n_{\mathbf{i},\nu}} \tag{30}$$

corresponds to the anharmonic lattice potential term $\hat{\mathcal{V}}_{anh}$ defined in Eq. (4). The third term

$$S_{disp}(x) = \Delta\tau \sum_{\substack{\mathbf{i},\nu \\ \mathbf{j},\gamma}} \sum_{\substack{n_{\mathbf{i},\nu} \\ n_{\mathbf{j},\gamma}}} \sum_{l} \frac{M_{n_{\mathbf{i},\nu}} M_{n_{\mathbf{j},\gamma}}}{M_{n_{\mathbf{i},\nu}} + M_{n_{\mathbf{j},\gamma}}} \left[ \tilde{\Omega}^2_{0,n_{\mathbf{i},\nu},n_{\mathbf{j},\gamma}} \left( x_{n_{l,\mathbf{i},\nu}} - x_{l,n_{\mathbf{j},\gamma}} \right)^2 + \frac{1}{12} \tilde{\Omega}^2_{a,n_{\mathbf{i},\nu},n_{\mathbf{j},\gamma}} \left( x_{l,n_{\mathbf{i},\nu}} - x_{l,n_{\mathbf{j},\gamma}} \right)^4 \right] \tag{31}$$

arises from dispersive couplings between QHO modes described by $\hat{\mathcal{U}}_{disp}$, defined in Eq. (5). The final term

$$S_{hol} = \begin{cases} -\frac{\Delta\tau}{2} \sum_{\sigma} \sum_{\substack{\mathbf{i},\nu \\ \mathbf{j},\gamma}} \sum_{n_{\mathbf{i},\nu}} \left( \tilde{\alpha}_{\sigma,1,n_{\mathbf{i},\nu},(\mathbf{j},\gamma)} x_{l,n_{\mathbf{i},\nu}} + \tilde{\alpha}_{\sigma,3,n_{\mathbf{i},\nu},(\mathbf{j},\gamma)} x^3_{l,n_{\mathbf{i},\nu}} \right) \\ 0 \end{cases} \tag{32}$$

arises due to the manner in which the Holstein-like interactions appearing in $\hat{\mathcal{V}}_{hol}$ are parameterized in Eq. (11).

The high-dimensional integral appearing in Eq. (24) is not analytically tractable, but lends itself to a numerical solution. In particular, DQMC simulations perform a Monte Carlo sampling of the HS fields $s$ and phonon fields $x$, using as Monte Carlo weights the argument of the integral in Eq. (24),

$$W(s,x) = e^{-S_B(x)} \prod_{\sigma=\uparrow,\downarrow} \det M_\sigma(\tau). \tag{33}$$

We note, however, that the Monte Carlo weights as determined by Eq. (33) are not strictly positive. They can take on both positive and negative values in many cases, and can become complex if other types of HS transformations are used. This aspect leads to the well-known Fermion sign problem [109–111]. To circumvent the negative weights, DQMC instead takes the absolute value of Eq. (33) as the Monte Carlo weights

$$\overline{W}(s,x) = |W(s,x)| = e^{-S_B(x)} \prod_{\sigma=\uparrow,\downarrow} |\det M_\sigma(\tau)|. \tag{34}$$

Specifically, applying the Metropolis-Hastings criteria, the acceptance probability for updating the field configuration from $(s, x)$ to $(s', x')$ is given by

$$P_{(s,x) \to (s',x')} = \min\left(1, \frac{\overline{W}(s', x')}{\overline{W}(s, x)}\right). \tag{35}$$

However, using $\overline{W}(s, x)$ necessitates employing a reweighting procedure to recover unbiased measurements from a simulation; this will be discussed more in Sec. 3.9.

## 3.2 DQMC Simulation Overview

This section briefly outlines the overarching structure of a DQMC simulation. By design, this structure closely mirrors how scripts using `SmoQyDQMC.jl` are written to perform simulations. We have provided numerous examples of such scripts in the online documentation, and Algorithm (1) provides an overview of what this structure might look like.

The goal of a DQMC simulation is to faithfully sample the relevant fields according to the probability distribution described by Eq. (34). In the case of `SmoQyDQMC.jl`, these fields correspond to either the Ising HS fields that result from decoupling the Hubbard interaction or the phonon fields. The fields are initialized to a random configuration at the beginning of a simulation. Then, updates to the field configurations are proposed using various methods, which are either accepted or rejected with a probability given by Eq. (35). Sec. 3.6 outlines an efficient procedure for updating the Ising HS fields, whereas sections 3.7 and 3.8 describe methods for efficiently updating the phonon fields.

Another important part of the simulation is making measurements as field configurations are sampled. However, measurements should not be performed at the start of a simulation as the initial field configuration is typically far from the target equilibrium distribution described by Eq. (34). Therefore, the fields are first updated $N_{\text{therm.}}$ times during an initial thermalization period to equilibrate the field configurations to the target distribution. The remainder of the simulation is then broken into $N_{\text{bin}}$ intervals, with $n_{\text{bin}}$ updates performed per interval, as shown in Algorithm (1). After each set of updates to the field configurations, measurements are made and recorded. Electronic correlation measurements are made using the single-particle electron Green's function (which is defined and discussed in Sec. 3.3). At the end of each interval, the average of the previous $n_{\text{bin}}$ measurements are written to the file. At the end of the simulation, these interval-averaged measurements are processed to generate final estimates for the expectation value of measured observables; the details of this analysis are discussed in Sec. 3.9. Algorithm (1) also shows how the chemical potential can be updated during a simulation to achieve a target electron density. Sec. 3.10 gives more details on this functionality.

Finally, the algorithms for updating fields and measuring electronic correlation functions require reliably calculating the single-particle electron Green's function matrices defined in Sec. 3.3. A naive approach to performing these calculations often results in numerical instabilities, rendering the result unreliable. Sec. 3.4 describes efficient algorithms for suppressing these numerical instabilities that would otherwise prevent the DQMC algorithm from functioning correctly.

## 3.3 Single-Particle Electron Green's Functions

A central quantity in DQMC simulations is the imaginary-time single-particle Green's function. This section provides a brief review of the definitions of this quantity and its properties.

---

**Algorithm 1** Overview of DQMC simulation structure

---

Initialize Ising HS fields $s$ and phonon fields $x$ to random initial configuration.

**for** $i \in [1, N_{\text{therm.}}]$ **do**

    Update Ising HS field $s$ using algorithm from Sec. 3.6.

    Update phonon fields $x$ using algorithms from sections 3.7 and 3.8.

    Update chemical potential; see Sec. 3.10

**end for**

**for** bin $\in [1, N_{\text{bins}}]$ **do**

    **for** $i \in [1, n_{\text{bin}}]$ **do**

        Update Ising HS field $s$ using algorithm from Sec. 3.6.

        Update phonon fields $x$ using algorithms from sections 3.7 and 3.8.

        Make measurements; see Sec. 3.3.

        Update chemical potential; see Sec. 3.10

    **end for**

    Write average value of measurements in current bin to file.

**end for**

Average binned measurements to get final estimates for measured observables; see Sec. 3.9.

---

The spin-$\sigma$ electron Green's function in the orbital basis is given by

$$
\begin{aligned}
\mathcal{G}_{\sigma,\mathbf{i},\mathbf{j}}^{\nu,\gamma}(\tau, \tau') &= \langle \hat{\mathcal{T}} \hat{c}_{\sigma,\mathbf{i},\nu}(\tau) \hat{c}_{\sigma,\mathbf{j},\gamma}^{\dagger}(\tau') \rangle \\
&= \begin{cases} \langle \hat{c}_{\sigma,\mathbf{i},\nu}(\tau) \hat{c}_{\sigma,\mathbf{j},\gamma}^{\dagger}(\tau') \rangle & \text{if } \tau \geq \tau' \\ -\langle \hat{c}_{\sigma,\mathbf{j},\gamma}^{\dagger}(\tau') \hat{c}_{\sigma,\mathbf{i},\nu}(\tau) \rangle & \text{if } \tau < \tau', \end{cases}
\end{aligned} \tag{36}
$$

where $\hat{\mathcal{T}}$ is the imaginary-time ordering operator. Eq. (36) describes the creation of a spin-$\sigma$ electron at orbital $\gamma$ in unit cell $\mathbf{j}$ at imaginary-time $\tau'$, and annihilation at orbital $\nu$ in unit cell $\mathbf{i}$ at imaginary time $\tau$. The imaginary-time axis spans the interval $0 \leq (\tau, \tau') < \beta$, and the electron Green's function is additionally subject to the aperiodic boundary condition

$$
\mathcal{G}_{\sigma,\mathbf{i},\mathbf{j}}^{\nu,\gamma}(\tau - \beta, \tau') = -\mathcal{G}_{\sigma,\mathbf{i},\mathbf{j}}^{\nu,\gamma}(\tau, \tau') \tag{37}
$$

when $\tau \neq \tau'$.

Given a fixed set of field configurations $s$ and $x$, the equal-time Green's can be related to the matrix elements of the Fermion determinant matrix by

$$
G_\sigma(\tau, \tau) = M_\sigma^{-1}(\tau) = \left[ I + B_\sigma(\tau, 0) B_\sigma(\beta, \tau) \right]^{-1}, \tag{38}
$$

where we have used the short-hand notation

$$
B_\sigma(\tau, \tau') = B_{\sigma,l} B_{\sigma,l-1} \ldots B_{l'+1}, \tag{39}
$$

with $B_\sigma(\tau, \tau - \Delta\tau) = B_{\sigma,l}$ and $B_\sigma(\tau, \tau) = I$. Therefore,

$$
G_\sigma(0, 0) = \left[ I + B_\sigma(\beta, 0) \right]^{-1} = \left[ I + B_{\sigma,L_\tau} B_{\sigma,L_\tau-1} \ldots B_{\sigma,1} \right]^{-1}, \tag{40}
$$

subject to the boundary condition

$$
G_\sigma(\beta, \beta) = G_\sigma(0, 0). \tag{41}
$$

The equal-time Green's function matrix $G_\sigma(\tau, \tau)$ can be propagated to adjacent imaginary-time slices in the forward and reverse directions using the relations

$$
G_\sigma(\tau + \Delta\tau, \tau + \Delta\tau) = B_{\sigma,l+1} G_\sigma(\tau, \tau) B_{\sigma,l+1}^{-1} \tag{42}
$$

and

$$G_\sigma(\tau - \Delta\tau, \tau - \Delta\tau) = B_{\sigma,l}^{-1} G_\sigma(\tau, \tau) B_{\sigma,l}, \tag{43}$$

respectively. These relationships between the Green's functions on adjacent time-slices are the basis for an efficient updating scheme when performing Metropolis-Hastings or Heat-bath sampling [2], as discussed in Sec. 3.6.

The time-displaced (or unequal-time) Green's functions $\mathcal{G}_{\sigma,\mathbf{i},\mathbf{j}}^{\nu,\gamma}(\tau, 0)$ and $\mathcal{G}_{\sigma,\mathbf{i},\mathbf{j}}^{\nu,\gamma}(0, \tau)$ correspond to the matrix elements of

$$G_\sigma(\tau, 0) = B_\sigma(\tau, 0) G_\sigma(0, 0) \tag{44a}$$
$$= [B_\sigma^{-1}(\tau, 0) + B_\sigma(\beta, \tau)]^{-1} \tag{44b}$$

and

$$G_\sigma(0, \tau) = G_\sigma(0, 0^+) B_\sigma^{-1}(\tau, 0)$$
$$= -[I - G_\sigma(0, 0)] B_\sigma^{-1}(\tau, 0) \tag{45a}$$
$$= -[B_\sigma^{-1}(\beta, \tau) + B_\sigma(\tau, 0)]^{-1}, \tag{45b}$$

respectively, where we have applied the boundary condition

$$G_\sigma(0, 0^+) = \lim_{\tau \to 0} G_\sigma(0, \tau) = -[I - G_\sigma(0, 0)]. \tag{46}$$

In similar fashion to the equal-time Green's function matrices, the time-displaced Green's function matrices can be propagated in the forward and reverse imaginary-time directions using the relations

$$G_\sigma(\tau, 0) = \begin{cases} B_\sigma(\tau, \tau') G_\sigma(\tau', 0) & \text{if } \beta > \tau > \tau' > 0 \\ B_\sigma^{-1}(\tau', \tau) G_\sigma(\tau', 0) & \text{if } \beta > \tau' > \tau > 0 \end{cases} \tag{47}$$

and

$$G_\sigma(0, \tau) = \begin{cases} G_\sigma(0, \tau') B_\sigma^{-1}(\tau, \tau') & \text{if } \beta > \tau > \tau' > 0 \\ G_\sigma(0, \tau') B_\sigma(\tau', \tau) & \text{if } \beta > \tau' > \tau > \beta. \end{cases} \tag{48}$$

Specific boundary conditions arise as a result of the aperiodic boundary condition set by Eq. (37), and the definition of the Fermionic Green's function in Eq. (36). In particular, the time-displaced Green's matrices satisfy

$$G_\sigma(\beta, 0) = \lim_{\tau \to \beta} G_\sigma(\tau, 0) = I - G_\sigma(0, 0) \tag{49}$$
$$G_\sigma(0, \beta) = \lim_{\tau \to \beta} G_\sigma(0, \tau) = -G_\sigma(0, 0), \tag{50}$$

where it is again assumed that $\tau > 0$.

Lastly, again assuming a fixed HS and phonon field configuration, higher order correlation functions can be measured by applying Wick's theorem to express them as sums of products of the single-particle electron Green's functions [12, 112].

## 3.4 Numerically Stable Framework for DQMC Simulations

The procedure for updating the HS ($s_{l,\mathbf{i},\nu}$) and phonon ($x_{l,n_{\mathbf{i},\nu}}$) fields requires calculating the equal-time Green's function matrices $G_\sigma(\tau, \tau)$ for all imaginary time slices $l \in [1, L_\tau]$. Similarly, performing measurements of any imaginary time-displaced correlation functions requires calculating the imaginary time-displaced Green's functions $G_\sigma(\tau, 0)$ and $G_\sigma(0, \tau)$.

A straightforward approach for computing these quantities is outlined in Algorithm 2. Unfortunately, this naive approach fails due to well-documented [2, 113–116] numerical instabilities associated with evaluating the ill-conditioned products of $B_{\sigma,l}$ matrices. Specifically, repeated matrix multiplication by the propagator matrices $B_{\sigma,l}$ accumulates numerical errors that quickly become severe. These numerical errors appear both when attempting to evaluate the $B_\sigma(\beta, 0)$ term appearing in Eq. (40), and also as a result of repeated applications of Eqs. (42), (47), and (48). The errors are then further amplified when matrix inversions are performed, as in Eq. (40).

Practical implementations of the DQMC algorithm have to overcome these numerical instabilities by introducing stable matrix factorizations [2, 113–116]. The `SmoQyDQMC.jl` package uses stabilization procedures based on those introduced in Ref. [114] and further discussed in Ref. [116]. Our package also stores intermediate matrix products to improve the algorithm's efficiency [5, 117, 118], an approach described in greater detail below. To outline this procedure, we first introduce the *LDR* matrix factorization, which represents products of propagator matrices $B_{\sigma,l}$ and is based on the column-pivoted *QR* factorization. For a non-singular square matrix *A*, its column-pivoted *QR* factorization is given by

$$AP = QR', \tag{51}$$

where $Q$ is a unitary matrix, $R'$ is an upper-triangular matrix, and $P$ is a permutation matrix. The corresponding $A = LDR$ factorization is then defined as product of matricies

$$L = Q, \tag{52a}$$

$$D = |\text{Diag}(R')|, \text{ and} \tag{52b}$$

$$R = |\text{Diag}(R')|^{-1}R'P^T. \tag{52c}$$

Here, both the unitary matrix *L* and matrix *R* are well-conditioned matrices, and *D* is a diagonal matrix. To further improve numerical stability, it is useful to factorize *D* as

$$D = D_{\max} D_{\min}, \tag{53}$$

where $D_{\max} = \max(D, 1)$ and $D_{\min} = \min(D, 1)$ [114].

Next, the imaginary-time axis of length $L_\tau$ is split into $N_s = L_\tau/n_s$ intervals of length $n_s$. The relative position within interval $n \in [1, N_s]$ is given by $l_n \in [1, n_s]$, where the corresponding imaginary-time slice is $l = l_n + (n-1)n_s$. For simplicity's sake, here we have assumed $\text{mod}(L_\tau, n_s) = 0$. While this will not be the case in general, it is relatively straightforward to generalize the algorithms outlined below to the case that this assumption does not hold.

---

**Algorithm 2** Numerically Unstable Forward Propagation Framework for DQMC Simulations

---

Calculate $G_\sigma(0, 0)$ using Eq. (40): $G_\sigma(0, 0) := [I + B_\sigma(\beta, 0)]^{-1}$
Initialize $G_\sigma(\tau, \tau)$: $G_\sigma(\tau, \tau) := G_\sigma(0, 0)$
Initialize $G_\sigma(\tau, 0)$: $G_\sigma(\tau, 0) := G_\sigma(0, 0)$
Initialize $G_\sigma(0, \tau)$ by applying Eq. (46): $G_\sigma(0, \tau) := -[I - G_\sigma(0, 0)]$
**for** $l \in 1, 2, \ldots, L_\tau$ **do**
    Apply Eq. (42): $G_\sigma(\tau, \tau) := B_{\sigma,l} G_\sigma(\tau - \Delta\tau, \tau - \Delta\tau) B_{\sigma,l}^{-1}$
    Apply Eq. (47): $G_\sigma(\tau, 0) := B_{\sigma,l} G_\sigma(\tau - \Delta\tau, 0)$
    Apply Eq. (48): $G_\sigma(0, \tau) := G_\sigma(0, \tau - \Delta\tau) B_{\sigma,l}^{-1}$

    [Insert Updates to Fields or Time-Displaced Correlation Measurements Here.]

**end for**

---

The composite propagator matrix associated with an interval $n$ is given by

$$\bar{B}_{\sigma,n} = B_{\sigma,nn_s} B_{\sigma,nn_s-1} \dots B_{\sigma,(n-1)n_s+1}$$
$$= B_\sigma(\Delta\tau\, n\, n_s,\ \Delta\tau\, (n-1)\, n_s), \tag{54}$$

with the product of $\bar{B}_{\sigma,n}$ matrices represented by

$$\bar{B}_{\sigma,n,n'} = \bar{B}_{\sigma,n}\bar{B}_{\sigma,n-1}\dots\bar{B}_{\sigma,n'+1}$$
$$= B_{\sigma,nn_s} B_{\sigma,nn_s-1}\dots B_{\sigma,n'n_s+1}$$
$$= B_\sigma(\Delta\tau\, n\, n_s,\ \Delta\tau\, n'\, n_s), \tag{55}$$

where $n > n'$ and $\bar{B}_{\sigma,n,n-1} = \bar{B}_{\sigma,n}$. The corresponding *LDR* factorization for $\bar{B}_{\sigma,n,n'}$ is denoted by

$$\bar{F}_{\sigma,n,n'} = \bar{L}_{\sigma,n,n'}\bar{D}_{\sigma,n,n'}\bar{R}_{\sigma,n,n'}. \tag{56}$$

Finally, $\bar{\mathbf{F}}_\sigma$ denotes an array of *LDR* factorizations of length $N_s$ to store sequential $\bar{F}_{\sigma,n,n'}$ factorizations.

With these definitions in hand, we now turn our attention to Algorithms (3a) and (3b) for propagating in imaginary-time the equal-time and time-displaced Green's function matrices in the forward ($\tau = 0 \rightarrow \tau = \beta$) and reverse ($\tau = \beta \rightarrow \tau = 0$) directions, respectively. In particular, note that the input state of that array $\bar{\mathbf{F}}$ in Algorithm (3b) matches the output state from Algorithm (3a). Likewise the input state of $\bar{\mathbf{F}}$ in Algorithm (3a) matches the output state from Algorithm (3b). Therefore, it is necessary to alternate the application of Algorithms (3a) and (3b). This results in a DQMC updating procedure that alternates sweeping forward and backward in imaginary time rather than cyclically. The advantage of this approach is that it allows the DQMC algorithm to retain a computational cost that scales linearly with $\beta$, while remaining numerically stable [117]. The parameter $n_s$ is now understood to be the period in imaginary-time with which the Green's function matrices are re-computed using a more expensive, but numerically stable procedure.

In practice, it is important to monitor the numerical stability of a simulation to determine whether $n_s$ needs to be decreased to increase numerical stability, or increased to reduce computational overhead. To accomplish this task, let $G_\sigma^{\text{stable}}(\tau,\tau)$ correspond to the equal-time Green's matrix that is recomputed using a numerically stable procedure, while $G_\sigma^{\text{naive}}(\tau,\tau)$ represents the same matrix, but generated via simple propagation in imaginary time using Eq. (42) or Eq. (43). We then keep track of the maximum element-wise difference

$$\delta G_\sigma = \max\left(|G_\sigma^{\text{stable}}(\tau,\tau) - G_\sigma^{\text{naive}}(\tau,\tau)|\right) \tag{57}$$

during a simulation. We would like $\delta G_\sigma$ remain below some maximum threshold $\delta G_\sigma < \delta G_{\text{max}}$, with $\delta G_{\text{max}} \approx 10^{-6}$ a sufficient upper bound in most cases. Typical values for the period of numerical stabilization that satisfy this stability condition are $n_s \sim 5-10$, but depend on the strength of the interactions, inverse temperature $\beta$, and $\Delta\tau$. When this condition is being violated in the course of a DQMC simulation, it is an indication that $n_s$ may need to be reduced. Conversely, if $\delta G_\sigma \ll \delta G_{\text{max}}$ throughout the simulation, then $n_s$ can often be increased to reduce the simulation's run time. `SmoQyDQMC.jl` offers useful functionality for reducing $n_s$ dynamically during a simulation if instances of $\delta G_\sigma > \delta G_{\text{max}}$ are detected too frequently. However, it is worth noting that DQMC simulations performed with `SmoQyDQMC.jl` retain a computational cost that scales linearly with $\beta$ for any $n_s$. Rather, reducing $n_s$ simply increases the computational prefactor associated with the linear dependence on $\beta$.

---

**Algorithm 3a** Numerically Stable Forward Propagation Framework for DQMC Simulations

---

**Input:** $G_\sigma(0,0)$

**Input:** $\bar{\mathbf{F}}_\sigma = \left[ \bar{F}_{\sigma,N_s,0} \, , \, \bar{F}_{\sigma,N_s,1} \, , \, \ldots \, , \, \bar{F}_{\sigma,N_s,N_s-1} \right]$

**for** $l \in 1, 2, \ldots, L_\tau - 1, L_\tau$ **do**

    Calculate $l \to (n, l_n)$

    Apply Eq. (42): $G_\sigma(\tau, \tau) := B_{\sigma,l} \, G_\sigma(\tau - \Delta\tau, \tau - \Delta\tau) \, B_{\sigma,l}^{-1}$

    Apply Eq. (47): $G_\sigma(\tau, 0) := B_{\sigma,l} \, G_\sigma(\tau - \Delta\tau, 0)$

    Apply Eq. (48): $G_\sigma(0, \tau) := G_\sigma(0, \tau - \Delta\tau) \, B_{\sigma,l}^{-1}$

    [Insert Updates to Fields or Time-Displaced Correlation Measurements Here.]

    **if** $l_n == n_s$ **then**

        Calculate $\bar{B}_{\sigma,n}$.

        **if** $n == 1$ **then**

            Calculate $\bar{F}_{\sigma,1,0} := \bar{B}_{\sigma,1}$

        **else**

            Retrieve $\bar{F}_{\sigma,n-1,0} = \bar{\mathbf{F}}_\sigma[n-1]$

            Calculate $\bar{F}_{\sigma,n,0} := \bar{B}_{\sigma,n}\bar{F}_{\sigma,n-1,0}$ using stabilization routine A.1

        **end if**

        Set $\bar{\mathbf{F}}_\sigma[n] := \bar{F}_{\sigma,n,0}$

        **if** $l == L_\tau$ **then**

            Evaluate Eq. (40) using routine A.4: $G_\sigma(0,0) := \left[ 1 + \bar{F}_{\sigma,N_s,0} \right]^{-1}$

            Evaluate Eq. (49): $G_\sigma(\beta,0) := I - G_\sigma(0,0)$

            Evaluate Eq. (50): $G_\sigma(0,\beta) := -G_\sigma(0,0)$

        **else**

            Retrieve $\bar{F}_{\sigma,N_s,n} = \bar{\mathbf{F}}_\sigma[n+1]$

            Evaluate Eq. (38) using routine A.3: $G_\sigma(\tau,\tau) := \left[ I + \bar{F}_{\sigma,n,0}\bar{F}_{\sigma,N_s,n} \right]^{-1}$

            Evaluate Eq. (44b) using routine A.5: $G_\sigma(\tau,0) := \left[ \bar{F}_{\sigma,n,0}^{-1} + \bar{F}_{\sigma,N_s,n} \right]^{-1}$

            Evaluate Eq. (45b) using routine A.5: $G_\sigma(0,\tau) := -\left[ \bar{F}_{\sigma,N_s,n}^{-1} + \bar{F}_{\sigma,n,0} \right]^{-1}$

        **end if**

    **end if**

**end for**

**Output:** $\bar{\mathbf{F}}_\sigma = \left[ \bar{F}_{\sigma,1,0} \, , \, \bar{F}_{\sigma,2,0} \, , \, \ldots \, , \, \bar{F}_{\sigma,N_s,0} \right]$

---

---

**Algorithm 3b** Numerically Stable Reverse Propagation Framework for DQMC Simulations

---

**Input:** $G_\sigma(0,0)$
**Input:** $\bar{\mathbf{F}}_\sigma = \left[ \bar{F}_{\sigma,1,0} , \bar{F}_{\sigma,2,0} , \dots , \bar{F}_{\sigma,N_s,0} \right]$
**for** $l \in L_\tau, L_\tau - 1, \dots, 2, 1$ **do**
    Calculate $l \rightarrow (n, l_n)$
    **if** $l == L_\tau$ **then**
        Apply Eq. (49): $G_\sigma(\beta, 0) := I - G_\sigma(0,0)$
        Apply Eq. (50): $G_\sigma(0, \beta) := -G_\sigma(0,0)$
    **else if** $l_n \neq n_s$ **then**
        Apply Eq. (43): $G_\sigma(\tau, \tau) := B_{\sigma,l+1}^{-1} G_\sigma(\tau + \Delta\tau, \tau + \Delta\tau) B_{\sigma,l+1}$
        Apply Eq. (47): $G_\sigma(\tau, 0) := B_{\sigma,l+1}^{-1} G_\sigma(\tau + \Delta\tau, 0)$
        Apply Eq. (48): $G_\sigma(0, \tau) := G_\sigma(0, \tau + \Delta\tau) B_{\sigma,l+1}$
    **end if**

    [Insert Updates to Fields or Time-Displaced Correlation Measurements Here.]

    **if** $l_n == 1$ **then**
        Calculate $\bar{B}_{\sigma,n}$.
        **if** $n == N_s$ **then**
            Calculate $\bar{F}_{\sigma,N_s,N_s-1} := \bar{B}_{\sigma,n}$
        **else**
            Retrieve $\bar{F}_{\sigma,N_s,n} = \bar{\mathbf{F}}_\sigma[n+1]$
            Calculate $\bar{F}_{\sigma,N_s,n-1} := \bar{F}_{\sigma,N_s,n} \bar{B}_{\sigma,n}$ using stabilization routine A.2
        **end if**
        Set $\bar{\mathbf{F}}_\sigma[n] := \bar{F}_{\sigma,N_s,n-1}$
        **if** $l == 1$ **then**
            Evaluate Eq. (40): $G_\sigma(0,0) := \left[ 1 + \bar{F}_{\sigma,N_s,0} \right]^{-1}$
            Evaluate Eq. (49): $G_\sigma(\beta, 0) := I - G_\sigma(0,0)$
            Evaluate Eq. (50): $G_\sigma(0, \beta) := -G_\sigma(0,0)$
        **else if** $l_n == 1$ **then**
            Retrieve $\bar{F}_{\sigma,n-1,0} = \bar{\mathbf{F}}_\sigma[n-1]$
            Evaluate Eq. (38) using routine A.4: $G_\sigma(\tau - \Delta\tau, \tau - \Delta\tau) := \left[ I + \bar{F}_{\sigma,n-1,0} \bar{F}_{\sigma,N_s,n-1} \right]^{-1}$
            Evaluate Eq. (44b) using routine A.5: $G_\sigma(\tau - \Delta\tau, 0) := \left[ \bar{F}_{\sigma,n-1,0}^{-1} + \bar{F}_{\sigma,N_s,n-1} \right]^{-1}$
            Evaluate Eq. (45b) using routine A.5: $G_\sigma(0, \tau - \Delta\tau) := -\left[ \bar{F}_{\sigma,N_s,n-1}^{-1} + \bar{F}_{\sigma,n-1,0} \right]^{-1}$
        **end if**
    **end if**
**end for**
**Output:** $\bar{\mathbf{F}}_\sigma = \left[ \bar{F}_{\sigma,N_s,0} , \bar{F}_{\sigma,N_s,1} , \dots , \bar{F}_{\sigma,N_s,N_s-1} \right]$

---

## 3.5 The Checkerboard Approximation

The `SmoQyDQMC.jl` package makes heavy use of the checkerboard approximation [119]. For example, this approximation is necessary for efficiently simulating optical SSH $e$-ph interactions, where the hopping integrals are modulated by the atomic displacements of phonon modes defined to live on the sites of the lattice [102]. Note that `SmoQyDQMC.jl` does allow users to run simulations without the checkerboard approximation, but this will slow the code down significantly.

The checkerboard approximation is motivated by the observation that the exact exponentiated kinetic energy matrices

$$\Gamma_{\sigma,l}(\Delta\tau) = e^{-\Delta\tau K_{\sigma,l}} \tag{58}$$

appearing in the definition of the propagator matrices $B_{\sigma,l}$ are dense $\mathcal{N} \times \mathcal{N}$ matrices. As a result, evaluating the product of $B_{\sigma,l}$ with another dense matrix scales as $O(\mathcal{N}^3)$, and constitutes one of the leading computational costs in DQMC simulations. Moreover, when SSH-like $e$-ph interactions are present (arising from the $\hat{\mathcal{K}}_{\mathrm{ssh}}$ term in the Hamiltonian), updating a phonon field $x_{l,n_{\mathrm{i},\nu}}$ necessitates diagonalizing the corresponding spin-$\sigma$ kinetic energy matrix $K_{\sigma,l}$ in order to re-exponentiate it. If this diagonalization is performed at every update, it would introduce an exorbitant additional computational cost and significantly slow the simulation. To ameliorate these computational hurdles, we introduce the order $O(\Delta\tau^2)$ checkerboard approximation, which replaces the $\Gamma_{\sigma,l}(\Delta\tau)$ dense matrices with sparse matrix representations. Applying this approximation reduces the cost of multiplying a dense matrix by $B_{\sigma,l}$ from $O(\mathcal{N}^3)$ to $O(\mathcal{N}^2)$. But more importantly, it reduces the cost of exponentiating the $K_{\sigma,l}$ matrices when a phonon field is updated from $O(\mathcal{N}^3)$ to $O(\mathcal{N})$.

Consider a spin-$\sigma$ kinetic energy matrix $K_{\sigma,(i,j)} = -t_{\sigma,i,j}$, where $t_{\sigma,i,j}$ is the total hopping amplitude associated with the bond connecting orbitals $i$ and $j$ in the lattice. This matrix can be expressed as a sum of bond matrices $K_\sigma = \sum_b k_{\sigma,b}$. Here, $b$ indexes each bond in the lattice, and the corresponding bond matrix is of the form

$$k_{\sigma,b} = \begin{bmatrix} \ddots & \vdots & & \vdots & \\ \cdots & 0 & \cdots & -t_{\sigma,i_b,j_b} & \cdots \\ & \vdots & \ddots & \vdots & \\ \cdots & -t^*_{\sigma,i_b,j_b} & \cdots & 0 & \cdots \\ & \vdots & & \vdots & \ddots \end{bmatrix}, \tag{59}$$

where a hopping amplitude may in general be complex, $t_{\sigma,i_b,j_b} = e^{i\phi_{\sigma,i_b,j_b}}|t_{\sigma,i_b,j_b}|$. For each bond matrix $k_{\sigma,b}$, a corresponding bond operator $\hat{k}_{\sigma,b} = -\sum_\sigma \left[ t_{\sigma,i_b,j_b}\hat{c}^\dagger_{\sigma,i_b}\hat{c}_{j_b} + \mathrm{h.c.} \right]$ can be defined. Exponentiating $k_{\sigma,b}$ results in

$$e^{-\Delta\tau k_{\sigma,b}} = \begin{bmatrix} 1 & & \vdots & & \vdots & \\ \cdots & \cosh(\Delta\tau|t_{\sigma,i_b,j_b}|) & \cdots & e^{i\phi_{\sigma,i_b,j_b}}\sinh(\Delta\tau|t_{\sigma,i_b,j_b}|) & \cdots \\ & \vdots & 1 & \vdots & \\ \cdots & e^{-i\phi_{\sigma,i_b,j_b}}\sinh(\Delta\tau|t_{\sigma,i_b,j_b}|) & \cdots & \cosh(\Delta\tau|t_{\sigma,i_b,j_b}|) & \cdots \\ & \vdots & & \vdots & 1 \end{bmatrix}. \tag{60}$$

Next, the bonds must be sorted into groups, or colors, such that the bonds of a given color do not overlap or touch. This condition corresponds to the bond operators $\hat{k}_b$ (and corresponding matrices $k_b$) of the same color all mutually commuting with one another.

The task of constructing these groups can be reduced to the edge coloring problem in graph theory. It is important that the minimum number of colors are used, as this improves

the accuracy of the checkerboard approximation. However, the precise composition of each color is not unique, and some coloring schemes are better than others. In this code, the colors are assigned by systematically iterating over the unit cells in the lattice, and assigning a color to each bond with a site contained in the current unit cell; for more information, we refer the reader to our `Checkerboard.jl` package [120].

Having assigned a color to each bond in the lattice, the total kinetic energy matrix may be expressed as

$$K_\sigma = \sum_{c=1}^{N_c} K_{\sigma,c} = \sum_{c=1}^{N_c} \left[ \sum_{b \in c} k_{\sigma,b} \right], \tag{61}$$

where $K_{\sigma,c}$ is the kinetic energy matrix associated with just the bonds $b$ assigned the color $c$, with $N_c$ the number of colors. Absent any approximation, the exponential of a single color matrix $K_{\sigma,c}$ is given by

$$\Gamma_{\sigma,c}(\Delta\tau) = e^{-\Delta\tau K_{\sigma,c}} = \prod_{b \in c} e^{-\Delta\tau k_{\sigma,b}}. \tag{62}$$

With these definitions in hand, the checkerboard approximation is given by

$$\Gamma_\sigma(\Delta\tau) \approx \tilde{\Gamma}_\sigma(\Delta\tau) + O(\Delta\tau^2), \tag{63}$$

where

$$\tilde{\Gamma}_\sigma(\Delta\tau) = \Gamma_{\sigma,N_c}(\Delta\tau)\dots\Gamma_{\sigma,c}(\Delta\tau)\dots\Gamma_{\sigma,1}(\Delta\tau) \tag{64}$$

is the checkerboard matrix. A simple and efficient method for multiplying a dense matrix by $\tilde{\Gamma}_\sigma(\Delta\tau)$ is given by Algorithm 2 in Ref. [119].

The efficacy of the checkerboard approximation deteriorates in models with longer range hopping. This occurs due to the increase in the average coordination number with distance, increasing the number of color groups the bonds need to be partitioned into. Moreover, while $\Gamma_\sigma(\Delta\tau)$ is Hermitian, the checkerboard matrix $\tilde{\Gamma}_\sigma(\Delta\tau)$ is not. In order to mitigate concerns regarding the accuracy of the checkerboard approximation, a symmeterized form $\tilde{\Gamma}_\sigma^\dagger(\frac{\Delta\tau}{2}) \cdot \tilde{\Gamma}_\sigma(\frac{\Delta\tau}{2})$ can be used instead. This form results in a Hermitian checkerboard matrix and significantly improves the overall accuracy of the checkerboard approximation [121]. In a DQMC simulation, this corresponds to using the checkerboard approximation in conjunction with the symmetric definition for the propagator matrices $B_{\sigma,l}$, as defined by Eq. (26).

`SmoQyDQMC.jl` has support for both the asymmetric or symmetric checkerboard approximation. When performing simulations models that include SSH interactions, it is recommended that the symmetric checkerboard approximation be used in order to help ensure the efficacy of the checkerboard approximation.

## 3.6 Efficient Local Updates of Hubbard-Stratonovich Fields

In this section, we outline a well-known method for efficiently updating the Ising HS fields introduced in Sec. 3.1 to decouple a local Hubbard interaction [2]. A naive, or brute-force, approach to updating the HS fields consists of re-calculating the fermion determinants from scratch each time an update to a field $s_{l,\mathbf{i},\nu}$ is proposed. Updating a single field would then scale as $O(\mathcal{N}^3)$, with the resulting computational cost to update all HS fields in space and imaginary time scaling as $O(\beta\mathcal{N}^4)$. It is possible, however, to reduce the cost of updating a single HS field from $O(\mathcal{N}^3)$ to $O(\mathcal{N}^2)$ by taking advantage of the fact that the $e^{-\Delta\tau V_{\sigma,l}}$ matrices that appear in the definition of the propagator matrices are diagonal [2]. This property allows one to reduce the computational complexity of updating all HS fields from $O(\beta\mathcal{N}^4)$ to $O(\beta\mathcal{N}^3)$. For a derivation of the method outlined in this section for updating Ising HS fields, and a discussion on how to define and update HS transformations for other types of fermionic interactions, we refer readers to Ref. [12] and Ref. [112].

We will assume an asymmetric form for the $B_{\sigma,l}$ matrices in the following discussion, as defined in Eq. (27). Given an update to a single HS field $(s_{l,\mathbf{i},\nu} \to s'_{l,\mathbf{i},\nu})$, the corresponding change in the propagator matrix may be expressed as

$$B'_{\sigma,l} = \left[I + \Delta_\sigma(\tau,i)\right]B_{\sigma,l}, \tag{65}$$

where

$$\Delta_\sigma(\tau,i) = \begin{bmatrix} 0 & & & & \\ & \ddots & & & \\ & & e^{\sigma\alpha\left(s'_{l,i}-s_{l,i}\right)} - 1 & & \\ & & & \ddots & \\ & & & & 0 \end{bmatrix} \tag{66}$$

is a matrix with a single non-zero matrix element at index $i$ on the diagonal. Employing the Sherman-Morrison matrix identity, one can show that the change in the corresponding fermion matrix determinant is given by the scalar equation

$$\begin{aligned} R_\sigma(s_{l,\mathbf{i},\nu} \to s'_{l,\mathbf{i},\nu}) &= \frac{\det G_\sigma(\tau,\tau)}{\det G'_\sigma(\tau,\tau)} \\ &= \det\left[I + \Delta_\sigma(\tau,i)(I - G_\sigma(\tau,\tau))\right] \\ &= 1 + \Delta_{\sigma,i,i}(\tau,i)(1 - G_{\sigma,i,i}(\tau,\tau)). \end{aligned} \tag{67}$$

The updated Green's function matrix can then be efficiently calculated from the old one using the rank-1 update

$$G'_\sigma(\tau,\tau) = G_\sigma(\tau,\tau)\left[I - R_\sigma^{-1}\Delta_\sigma(\tau,i)(I - G_\sigma(\tau,\tau))\right], \tag{68}$$

which in terms of matrix elements is

$$G'_{\sigma,j,k}(\tau,\tau) = G_{\sigma,j,k}(\tau,\tau) - R_\sigma^{-1}G_{\sigma,j,i}(\tau,\tau)\Delta_{\sigma,i,i}(\tau,i)\left[\delta_{i,k} - G_{\sigma,i,k}(\tau,\tau)\right]. \tag{69}$$

Additional gains in efficiency can also be made by accumulating these updates in a delayed updating scheme [122].

This local updating scheme forms the basis for efficiently sampling HS fields across all spatial sites and imaginary times. One begins by proposing and accepting/rejecting updates to the HS fields for a single time slice, requiring $O(\mathcal{N}^3)$ operations. After all of the updates have been proposed for this imaginary time, the Green's function is advanced to the next time slice using Eq. (42), and the process is repeated. In practice, the imaginary time axis is sequentially iterated over, and Eq. (67) and Eq. (69) are used to efficiently update the HS fields at each imaginary time slice. While algorithms (3a) and (3b) are used to iterate over the imaginary time slices, the order the HS fields are iterated over at each time-slice is randomized to help reduce autocorrelation times. Note that a similar approach may be used to sample the fields when the symmetric definition for the $B_{\sigma,l}$ matrices [Eq. (26)] is used. However, in this case the Green's function matrix appearing in eqs. (67) to (69) is replaced by

$$\tilde{G}_\sigma(\tau,\tau) = e^{\frac{\Delta\tau}{2}K_l} G_\sigma(\tau,\tau) e^{-\frac{\Delta\tau}{2}K_l}. \tag{70}$$

Finally, we note that the same updating scheme can be used to sample the phonon fields in certain types of electron-phonon models (e.g. the Holstein model). However, it is not straightforward, or even necessarily possible, to formulate an efficient local updating scheme for arbitrary types of electron-phonon interactions. Moreover, even when local updates can be used they are inefficient in de-correlating the phonon fields, particularly when the phonon

energy is much smaller than the electronic hopping ($\Omega/t \lesssim 0.5$) [67, 123, 124]. Various alternative sampling methods have been proposed, including self-learning Monte Carlo [123, 124] and Langevin and HMC methods [92, 125–127], which can reduce the autocorrelation time associated with sampling the phonon fields. `SmoQyDQMC.jl` uses an optimized HMC method for sampling the phonon fields, as outlined in the next section.

### 3.7 HMC Updates of Phonon Fields

In the `SmoQyDQMC.jl` package, specialized hybrid Monte Carlo (HMC) updates are used to sample the phonon fields in DQMC simulations of $e$-ph models. The HMC method, also frequently referred to as Hamiltonian Monte Carlo, was first developed by the lattice gauge theory community [92], and has since become a widely used tool for sampling continuous random variables more broadly [127]. In an HMC update, the phonon fields evolve according to a fictitious Hamiltonian dynamics to construct proposed global updates to every phonon field simultaneously.

To define the fictitious Hamiltonian dynamics used to evolve the phonon fields, the DQMC Monte Carlo weight defined in Eq. (34) is re-expressed as

$$\overline{W}(s, x) = e^{-S(s,x)}, \tag{71}$$

where

$$S(s, x) = S_{\mathrm{B}}(x) + S_{\mathrm{F}}(s, x) \tag{72}$$

defines an effective action. The first term $S_{\mathrm{B}}(x)$, given by Eq. (28), is the purely bosonic contribution to the total action. The second term $S_{\mathrm{F}}(s, x)$ describes the fermionic contribution, and is given by

$$S_{\mathrm{F}}(s, x) = \sum_{\sigma} S_{\mathrm{F},\sigma}(s, x) = \sum_{\sigma} \mathrm{Tr} \log |G_{\sigma}(\tau, \tau)|, \tag{73}$$

valid for all $\tau$. Moving forward, we shall suppress reference to the HS fields $s$, as they will be treated as a constant while proposing updates to the phonon fields $x$.

Next, a conjugate momentum $p$ is introduced for each phonon field $x$, which allows us to define the effective Hamiltonian

$$H(x, p) = S(x) + \frac{1}{2} p^T \mathcal{M}^{-1} p, \tag{74}$$

which may be interpreted as the sum of "potential" and "kinetic" energies, where $\mathcal{M}$ is a positive-definite dynamical mass matrix. The corresponding Hamiltonian equations of motion are

$$\begin{aligned} \dot{p} &= -\frac{\partial H}{\partial x} = -\frac{\partial S}{\partial x} \\ \dot{x} &= \frac{\partial H}{\partial p} = \mathcal{M}^{-1} p, \end{aligned} \tag{75}$$

defining a symplectic, time-reversible, and energy-conserving dynamics.

To perform an HMC update, the first step is to directly sample the momentum according to the equilibrium Boltzmann distribution $\exp(-p^T \mathcal{M}^{-1} p / 2)$ according to

$$p = \sqrt{\mathcal{M}} R, \tag{76}$$

where each element of the random vector $R$ is drawn from a standard normal distribution. In the simplest approach, the dynamical mass matrix is set to the identity, $\mathcal{M} = I$. Next, the

Hamiltonian dynamics are evolved for $N_t$ time-steps using the leapfrog integration method

$$
\begin{aligned}
p_{t+1/2} &= p_t + \frac{\Delta t}{2} f_t \\
x_{t+1} &= x_t + \Delta t \mathcal{M}^{-1} p_{t+1/2} \\
p_{t+1} &= p_{t+1/2} + \frac{\Delta t}{2} f_{t+1},
\end{aligned}
\tag{77}
$$

where $\Delta t$ is the integration step size and

$$
f_t = -\frac{\partial S}{\partial x_t}
\tag{78}
$$

is the force driving the dynamics. Like the underlying Hamiltonian dynamics, the leapfrog integration method is symplectic and time-reversible. As a result, absent numerical instabilities, the total energy $H(x)$ will be conserved to $O(\Delta t^2)$ for arbitrarily long trajectories. Lastly, the final state $(x_f, p_f)$ of the HMC trajectory replaces the initial state $(x_i, p_i)$ with a probability given by the Metropolis-Hastings criteria

$$
P = \min\left(1, e^{-\Delta H}\right),
\tag{79}
$$

where $\Delta H = H(x_f, p_f) - H(x_i, p_i)$. Crucially, the HMC method exactly satisfies detailed balance as a result of the leapfrog integration method being time-reversible and symplectic.

The most expensive part of performing an HMC update is evaluating the derivative of the action

$$
\frac{\partial S}{\partial x} = \frac{\partial S_B}{\partial x} + \frac{\partial S_F}{\partial x}.
\tag{80}
$$

More specifically, evaluating the fermionic contribution to the derivative $\frac{\partial S_F}{\partial x}$ is the dominant cost, scaling as $O(\beta \mathcal{N}^3)$; Sec. 3.7.1 discusses how to evaluate this derivative.

Unfortunately, applying the basic HMC method outlined in this section still results in long autocorrelation times that stem from the disparate timescales introduced to the Hamiltonian dynamics by the bosonic action $S_B(x)$. Sec. 3.7.2 introduces a refined HMC method that utilizes two complementary methods for addressing this issue. However, while HMC updates are a powerful method for sampling phonon fields, Sec. 3.8 discusses intermittently supplementing the HMC updates with other types of global updates to help further reduce autocorrelation times and mitigate ergodicity concerns in certain situations. Lastly, while one might consider decoupling the Hubbard interaction using continuous HS fields which could then be sampled with HMC updates, the absence of a bosonic contribution coupling those fields in the imaginary time direction is known to make the approach ineffective [91,93].

### 3.7.1 Evaluating the Derivative of the Action

The most computationally expensive aspect of performing a HMC update is calculating the derivative of the action at each time step during the HMC trajectory. Evaluating the derivative of the bosonic action $S_B(x)$ is straightforward and fast to evaluate, with a computational cost that scales as $O(\beta \mathcal{N})$.

Taking the derivative of the fermionic action for a single spin species $S_F(x)$ is both more involved, and more computationally expensive, scaling as $O(\beta \mathcal{N}^3)$. The derivative of the fermionic action for just a single spin species $S_{F,\sigma}(x)$, as defined in Eq. (73), is

$$
\frac{\partial S_{F,\sigma}}{\partial x_{l,n}} = \mathrm{Tr}\left[\frac{\partial B_{\sigma,l}}{\partial x_{l,n}} B_{\sigma,l}^{-1}\left(G_\sigma(\tau, \tau) - I\right)\right],
\tag{81}
$$

where $l$ is the imaginary time slice and $n$ specifies the phonon mode.

In the case that the propagator matrices $B_{\sigma,l}$ are defined using the asymmetric definition given in Eq. (27), then Eq. (81) becomes

$$\frac{\partial S_{\text{F},\sigma}}{\partial x_{l,n}} = \text{Tr}\left[\frac{\partial \Lambda_{\sigma,l}}{\partial x_{l,n}}\Lambda_{\sigma,l}^{-1}\big(G_\sigma(\tau,\tau)-I\big)\right] + \text{Tr}\left[\frac{\partial \Gamma_{\sigma,l}}{\partial x_{l,n}}\Gamma_{\sigma,l}^{-1}\big(\Lambda_l^{-1}G_\sigma(\tau,\tau)\Lambda_l - I\big)\right], \tag{82}$$

where $\Lambda_{\sigma,l} = \exp(-\Delta\tau V_{\sigma,l})$ and $\Gamma_{\sigma,l} = \exp(-\Delta\tau K_{\sigma,l})$. If the $B_{\sigma,l}$ matrices are instead defined using the symmetric definition given in (26), then Eq. (81) instead becomes

$$\begin{aligned}
\frac{\partial S_{\text{F},\sigma}}{\partial x_{l,n}} &= \text{Tr}\left[\frac{\partial \Gamma_{\sigma,l}}{\partial x_{l,n}}\Gamma_{\sigma,l}^{-1}\big(G_\sigma(\tau,\tau)-I\big)\right] + \text{Tr}\left[\frac{\partial \Lambda_{\sigma,l}}{\partial x_{l,n}}\Lambda_{\sigma,l}^{-1}\big(\Gamma_{\sigma,l}^{-1}G_\sigma(\tau,\tau)\Gamma_{\sigma,l} - I\big)\right] \\
&\quad + \text{Tr}\left[\frac{\partial \Gamma_{\sigma,l}^\dagger}{\partial x_{l,n}}\Gamma_{\sigma,l}^{-\dagger}\big(\Lambda_{\sigma,l}^{-1}\Gamma_{\sigma,l}^{-1}G_\sigma(\tau,\tau)\Gamma_{\sigma,l}\Lambda_{\sigma,l} - I\big)\right],
\end{aligned} \tag{83}$$

where $\Lambda_{\sigma,l} = \exp(-\Delta\tau V_{\sigma,l})$ and $\Gamma_{\sigma,l} = \exp(-\Delta\tau K_l/2)$.

At this point, the simplification

$$\text{Tr}\left[\frac{\partial \Lambda_{\sigma,l}}{\partial x_{l,n}}\Lambda_{\sigma,l}^{-1}A\right] = -\Delta\tau\,\text{Tr}\left[\frac{\partial V_{\sigma,l}}{\partial x_{l,n}}A\right] \tag{84}$$

can be applied to equations (82) and (83), where $A$ is an $\mathcal{N}\times\mathcal{N}$ matrix. Also, when the exact form for the exponentiated kinetic energy matrix is used, and not the checkerboard approximation, then the substitution

$$\text{Tr}\left[\frac{\partial \Gamma_{\sigma,l}}{\partial x_{l,n}}\Gamma_{\sigma,l}^{-1}A\right] = -\Delta\tau\,\text{Tr}\left[\frac{\partial K_{\sigma,l}}{\partial x_{l,n}}A\right] \tag{85}$$

can also be applied, and $\Gamma_{\sigma,l}^\dagger = \Gamma_{\sigma,l}$. However, if the checkerboard approximation is being used, then Eq. (85) is no longer correct. Rather, applying the chain rule and taking advantage of the cyclic property of the trace, the relationship

$$\text{Tr}\left[\frac{\partial \tilde{\Gamma}_l}{\partial x_{l,n}}\tilde{\Gamma}_l^{-1}A\right] = -\Delta\tau\sum_{c=1}^{N_c}\left(\text{Tr}\left[\frac{\partial K_{\sigma,l,c}}{\partial x_{l,n}}\big(\Gamma_{l,c+1}^{-1}\dots\Gamma_{l,N_c}^{-1}\,A\,\Gamma_{l,N_c}\dots\Gamma_{c+1}\big)\right]\right), \tag{86}$$

should be used instead, where $\tilde{\Gamma}_l$ is the checkerboard approximation for $\Gamma_{\sigma,l}$ as defined in Eq. (64). In similar fashion, the simplification

$$\text{Tr}\left[\frac{\partial \tilde{\Gamma}_l^\dagger}{\partial x_{l,n}}\tilde{\Gamma}_l^{-\dagger}A\right] = -\Delta\tau\sum_{c=1}^{N_c}\left(\text{Tr}\left[\frac{\partial K_{\sigma,l,c}}{\partial x_{l,n}}\big(\Gamma_{l,c-1}^{-1}\dots\Gamma_{l,1}^{-1}\,A\,\Gamma_{l,1}\dots\Gamma_{c-1}\big)\right]\right) \tag{87}$$

can be applied as well.

In practice, to evaluate the derivative of the action we sequentially iterate over the imaginary time axis using Algorithm (3a) or (3b), generating each equal-time Green's function matrix $G_\sigma(\tau,\tau)$. This matrix is then used to calculate the derivative of the action with respect to the phonon fields for the current imaginary time slice. Doing this for each imaginary time slice, we evaluate the derivative of the action with respect to all the phonon fields while retaining $O(\beta\mathcal{N}^3)$ scaling in the DQMC simulations.

### 3.7.2 Resolving Disparate Timescales in the Bosonic Action

In this section, we consider an isolated QHO with mass $M$; generalizing this discussion to a collection of QHOs is straightforward.

An important cause of long autocorrelation times in DQMC simulations of $e$-ph models is the QHO action $S_{\text{qho}}(x)$, defined in Eq. (29). The action associated with a single QHO is

$$S_{\text{qho}}(x) = \Delta\tau \sum_l \left[ \frac{M\Omega^2}{2} x_l^2 + \frac{M}{2} \left( \frac{x_{l+1} - x_l}{\Delta\tau} \right)^2 \right], \tag{88}$$

with the resulting the resulting forces in a corresponding Hamiltonian dynamics given by

$$f_{\text{qho},l} = -\frac{\partial S_{\text{qho}}}{\partial x_l} = -\Delta\tau M \left[ \Omega^2 x_l + (2x_l - x_{l+1} - x_{l-1}) \right]. \tag{89}$$

Next, defining the discrete Fourier transform in imaginary time as

$$\tilde{f}_n = \mathcal{F} \cdot f_l = \frac{1}{\sqrt{L_\tau}} \sum_l e^{-i\frac{2\pi}{L_\tau} nl} f_l = \frac{1}{\sqrt{L_\tau}} \sum_l e^{-i\tau\omega_n} f_l, \tag{90}$$

and applying it to $f_{\text{qho},l}$ results in

$$\tilde{f}_{\text{qho},n} = -M\Delta\tau\Omega^2 \left[ 1 + \frac{4}{\Delta\tau^2\Omega^2} \sin^2\left( \frac{\pi n}{L_\tau} \right) \right] \tilde{x}_n, \tag{91}$$

for $n \in [0, L_\tau)$ and $\tau = l \cdot \Delta\tau$. The dynamical modes $\tilde{x}_n$ are the Fourier transform of the phonon fields in imaginary time $x_l$.

The resulting Hamiltonian equations of motion in frequency space associated with $S_{\text{qho}}(x)$ are given by

$$\begin{aligned} \dot{\tilde{p}}_n &= -\tilde{k}_n\,\tilde{x}_n \\ \dot{\tilde{x}}_n &= \tilde{M}_n^{-1}\,\tilde{p}_n, \end{aligned} \tag{92}$$

which describes a system of $L_\tau$ independent harmonic oscillators with spring constants

$$\tilde{k}_n = \Delta\tau M\Omega^2 \left[ 1 + \frac{4}{\Delta\tau^2\Omega^2} \sin^2\left( \frac{\pi n}{L_\tau} \right) \right]. \tag{93}$$

The resulting ratio of the magnitude of the forces for the fastest ($\tilde{x}_{L_\tau/2}$) and slowest ($\tilde{x}_0$) dynamical modes is

$$\frac{\tilde{k}_{L_\tau/2}}{\tilde{k}_0} = \left[ 1 + \frac{4}{\Delta\tau^2\Omega^2} \right] \gg 1, \tag{94}$$

demonstrating that $S_{\text{qho}}(x)$ introduces disparate timescales to the dynamics, especially as one must choose $\Delta\tau$ small to preserve the accuracy of the Trotter approximation.

This results in standard HMC updates needing to use a small integration time-step $\Delta t$ to resolve the high frequency dynamical modes, resulting in long autocorrelation times for the low frequency dynamical modes. Similarly, when performing local updates, in order to attain a reasonable acceptance rate, the proposed changes to the phonon field needs needs to be very small relative to the QHO characteristic length scale $\Delta X = 1/\sqrt{2M\Omega}$, once again giving rise to long autocorrelation times.

One successful approach for addressing this issue is the Fourier acceleration method, whereby carefully selected values for the dynamical mass $\tilde{M}_n$ appearing in Eq. (92) are selected so as to reduce autocorrelation times [59, 125, 126, 128, 129]. Specifically, the dynamically masses are given by

$$\tilde{M}_n = \Delta\tau M \left[ 1 + \frac{4}{\Delta\tau^2\Omega_{\text{reg}}^2} \sin^2\left( \frac{\pi n}{L_\tau} \right) \right], \tag{95}$$

---

**Algorithm 4** EFA

> **Inputs:** $x(t)$, $p(t)$, $\Delta t$
> Calculate regularized harmonic frequency: $\Omega_{\text{reg}} = \sqrt{1 + \eta_{\text{reg}}^2}\,\Omega$
> Fourier transform phonon fields: $\tilde{x}(t) = \mathcal{F} \cdot x(t)$
> Fourier transform conjugate momentum: $\tilde{p}(t) = \mathcal{F} \cdot p(t)$
> **for** $n \in [0, L_\tau - 1]$ **do**
> $\quad$ Calculate dynamical spring constant: $\tilde{k}_n = \Delta\tau M \Omega^2 \left[1 + \frac{4}{\Delta\tau^2 \Omega^2} \sin^2\left(\frac{\pi n}{L_\tau}\right)\right]$
> $\quad$ Calculate dynamics mass: $\tilde{M}_n = \Delta\tau M \left[1 + \frac{4}{\Delta\tau^2 \Omega_{\text{reg}}^2} \sin^2\left(\frac{\pi n}{L_\tau}\right)\right]$
> $\quad$ Calculate dynamical harmonic frequency: $\tilde{\omega}_n = \sqrt{\tilde{k}_n / \tilde{M}_n}$
> $\quad$ Evolve phonon fields: $\tilde{x}_n(t + \Delta t) = \tilde{x}_n(t)\,\cos(\tilde{\omega}_n \Delta t) + \tilde{p}_n\,\sin(\tilde{\omega}_n \Delta t) / (\tilde{M}_n \tilde{\omega}_n)$
> $\quad$ Evolve conjugate momentum: $\tilde{p}_n(t + \Delta t) = \tilde{p}_n(t)\,\cos(\tilde{\omega}_n \Delta t) - \tilde{x}_n\,\sin(\tilde{\omega}_n \Delta t) \times (\tilde{M}_n \tilde{\omega}_n)$
> **end for**
> Inverse Fourier transform conjugate momentum: $p(t + \Delta t) = \mathcal{F}^\dagger \cdot \tilde{p}(t + \Delta t)$
> Inverse Fourier transform phonon fields: $x(t + \Delta t) = \mathcal{F}^\dagger \cdot \tilde{x}(t + \Delta t)$
> **Output:** $x(t + \Delta t)$, $p(t + \Delta t)$

---

where $\Omega_{\text{reg}} = \sqrt{1 + \eta_{\text{reg}}^2}\,\Omega$ and $\eta_{\text{reg}} \in \mathbb{R}_{\geq 0}$ acts as a regularization parameter. Transforming back to imaginary time, this corresponds to a dynamical mass matrix, as appears in Eq. (75), given by

$$\mathcal{M} = \mathcal{F}^\dagger \tilde{M} \mathcal{F}, \tag{96}$$

where $\tilde{M}$ is a diagonal matrix with the values along the diagonal given by Eq. (95). In the case that $\eta_{\text{reg}} = 0$, the dynamical frequencies $\tilde{\omega}_n = \sqrt{\tilde{k}_n / \tilde{M}_n}$ for the harmonic oscillators described by Eq. (92) all equal the QHO frequency $\Omega$, resulting in all the dynamical modes $\tilde{x}_n$ evolving at the same rate. In opposite limit that $\eta_{\text{reg}} = \infty$, the dynamical mass matrix simply reduces to the scalar value $\mathcal{M} = \Delta\tau M$. Intuitively, decreasing $\eta_{\text{reg}}$ from infinity to zero should be thought of as increasing the mass of the high-frequency dynamical modes, thereby slowing them down so that they evolve at the same rate as the low-frequency dynamical modes.

In `SmoQyDQMC.jl` we use a slightly modified version of the exact Fourier acceleration hybrid Monte Carlo (EFA-HMC) method that was recently introduced in Ref. [130]. This approach takes advantage of the fact that the equations of motion in Eq. (92) are analytically integrable using the solution

$$\begin{aligned}
\tilde{x}_n(t) &= \tilde{x}_n(0)\,\cos(\tilde{\omega}_n t) + \tilde{p}_n(0)\,\sin(\tilde{\omega}_n t) / (\tilde{M}_n \tilde{\omega}_n) \\
\tilde{p}_n(t) &= \tilde{p}_n(0)\,\cos(\tilde{\omega}_n t) - \tilde{x}_n(0)\,\sin(\tilde{\omega}_n t) \times (\tilde{M}_n \tilde{\omega}_n),
\end{aligned} \tag{97}$$

given the initial conditions $\tilde{x}_n(0)$ and $\tilde{p}_n(0)$ for each dynamical mode. This analytic integration of the phonon fields and conjugate momentum, referred to as exact Fourier acceleration (EFA), is outlined in Algorithm (4). The full EFA-HMC method used in `SmoQyDQMC.jl` is then presented in Algorithm (5). An important detail in Algorithm (5) is that the time-step is randomized at the start of each HMC trajectory, with the amount of randomization controlled by the parameter $\delta \in (-1, 1)$. This practice helps avoid ergodicity concerns that can arise as the result of quasi-periodic behavior, an issue that can occur when the electrons only weakly couple to the high frequency modes in the dynamics [131].

It should be noted that that when $\eta_{\text{reg}} = \infty$ and $\delta = 0$, Algorithm (5) becomes equivalent to the version of the EFA-HMC method originally introduced in Ref. [130]. In practice, a good starting place for performing EFA-HMC updates is to set $N_t \cdot \Delta t \approx \pi/(2\Omega)$, with $N_t = 4$, $\eta_{\text{reg}} = 0.0$ and $\delta = 0.05$ [132]. If the acceptance rate is low ($\lesssim 60\%$), the first thing to try is decreasing $\Delta t$ and increasing $N_t$ such that their product remains constant. If the acceptance

---

**Algorithm 5** EFA-HMC

---

> **Input:** $x$, $\Delta t$, $N_t$, $\delta$
> Randomize time-step: $\Delta t := [1 + \text{Uniform}(-\delta, \delta)] \times \Delta t$
> Record initial phonon configuration: $x_i := x$
> Sample a vector of standard normal random numbers: $R \sim N(0, 1)$
> Initialize momentum $p$ using Eq. (76): $p := \sqrt{\mathcal{M}} R$
> Calculate initial energy: $H_i := S(x_i) + \frac{1}{2} p^T \mathcal{M} p$
> **for** $t \in [1, Nt]$ **do**
> > Apply algorithm (4): $(x, p) := \text{EFA}(x, p, \Delta t/2)$
> > Calculate relevant force: $f := -\left( \frac{\partial S}{\partial x} - \frac{\partial S_{\text{qho}}}{\partial x} \right)$
> > Update the momentum: $p := p + \Delta t \cdot f$
> > Apply algorithm (4): $(x, p) := \text{EFA}(x, p, \Delta t/2)$
> **end for**
> Calculate final energy: $H_f := S(x_f) + \frac{1}{2} p_f^T \mathcal{M} p_f$
> Calculate change in energy: $\Delta H := H_f - H_i$
> Calculate acceptance probaility: $P := \min(1, e^{-\Delta H})$
> Sample $r \sim \text{Uniform}(0, 1)$
> **if** $r < P$ **then**
> > Accept final phonon configuration: $x := x_f$
> **else**
> > Revert to initial phonon configuration: $x := x_i$
> **end if**
> **Output:** $x$

---

rate is very large ($\gtrsim 95\%$) it may be worth decreasing $N_t$ while increasing $\Delta t$, or increasing $\eta_{\text{reg}}$.

## 3.8 Reflection and Swap Updates

While HMC updates help decorrelate the phonon fields, other factors can increase the auto-correlation time. In the case of the Holstein model, the $e$-ph interaction induces an effective phonon mediated electron-electron attraction, giving rise to heavy bipolaron physics at moderate to large coupling, also resulting in long autocorrelation times. Likewise, DQMC simulations of the repulsive Hubbard with strong interactions ($U/t \gg 1$) contend with low acceptance rates for local updates as the HS fields develop a tendency to align in the imaginary time direction [133].

Another potential issue that can arise is a formal ergodicity problem if only HMC updates are used to sample the phonon fields. In particular, the Fermion determinant $\det M_\sigma$ going to zero corresponds to the action $S(x)$ diverging. Thus contours of $\det M_\sigma = 0$ in the phase space of phonon configurations describe nodal surfaces that separate regions of $\det M_\sigma > 0$ and $\det M_\sigma < 0$ by an infinite potential energy barrier $S(x) \to \infty$. A typical HMC update cannot cross these surfaces [93, 134], which introduces a formal ergodicity problem that is both difficult to predict *a priori* and challenging to diagnose. There is no general guarantee that prevents this from occurring, and has been observed in simulations of $e$-ph models in the anti-adiabatic limit ($\Omega/t \geq 1$) [129].

For these reasons, `SmoQyDQMC.jl` includes two additional types of updates, termed reflection and swap updates. In the case of a reflection update, the phonon fields of a randomly chosen phonon mode in the lattice are reflected about the origin for all imaginary times simultaneously. In a swap update, two phonon modes in the lattice are randomly chosen, and their phonon fields are interchanged, or "swapped", for all imaginary-time slices [126]. Both these

updates can also be used with Ising HS fields in the same way. The utility of these types of updates lies in the fact that they propose large, non-continuous changes to multiple degrees of freedom simultaneously, allowing simulations to cross regions of phase space that would otherwise be inaccessible using smaller, incremental updates.

## 3.9 Error Estimation and Reweighting

This section will review how `SmoQyDQMC.jl` computes the error associated with measured observables using the binning method and how the sign problem is addressed with reweighting [135] and the Jackknife algorithm [12].

To reliably calculate the error associated with the sample mean for a measured observable, effectively independent samples are required. However, the sequence of states generated by Markov chain Monte Carlo (MCMC) algorithms like DQMC are highly correlated. A standard approach to addressing this issue is the binning, or blocking method [12]. In this approach, the sequence of measurements generated in a MCMC simulation is partitioned into equally sized bins, or intervals, and the average value is computed for each bin. Once the bins become sufficiently large, containing a number of sequential measurements larger than the autocorrelation time, the average values associated with each bin may be treated as statistically independent samples. To calculate the error, one then calculates the sample standard deviation of the mean associated with the binned averages.

A `SmoQyDQMC.jl` DQMC simulation is structured such that $N_{\text{meas}}$ measurements are made during the simulation, which are aggregated and written to binary file $N_{\text{bins}}$ times during the simulation using the `JLD2.jl` package [136]. Therefore, each set of measurements written to file is the average of $N_{\text{binsize}} = (N_{\text{meas}}/N_{\text{bins}})$ individual measurements, where it is assumed that $\text{mod}(N_{\text{meas}}, N_{\text{bins}}) = 0$. Once the simulation is complete, and as long as the binary data files persist, the mean and error for any measurement can be calculated using $n_{\text{bins}}$ bins, where $n_{\text{bins}} \leq N_{\text{bins}}$ and $\text{mod}(N_{\text{bins}}, n_{\text{bins}}) = 0$.

The binning method outlined above is a generic method for calculating the error associated with estimates generated by a MCMC simulation. However, the situation in a generic DQMC simulation is more complicated. The expectation value of an observable $\hat{O}$ is given by

$$\langle O \rangle = \frac{\text{Tr}\left[ e^{-\beta\hat{\mathcal{H}}} \, \hat{O} \right]}{\text{Tr}\left[ e^{-\beta\hat{\mathcal{H}}} \right]}, \tag{98}$$

which, in the context of a DQMC simulation, is reformulated as

$$\langle O \rangle = \frac{\sum_s \int \mathcal{D}x \, W(s,x) \, O}{\sum_s \int \mathcal{D}x \, W(s,x)}, \tag{99}$$

where $W(s,x)$ is defined in Eq. (33). Ideally, this expression would be directly evaluated by performing a MCMC simulation with $W(s,x)$ used as the Monte Carlo weights. Unfortunately, the sign problem prevents this direct approach, which manifests as $W(s,x)$ not remaining a strictly positive real number. As a result, `SmoQyDQMC.jl` instead uses $\overline{W}(s,x)$ as the Monte Carlo weight in DQMC simulations, as defined in Eq. (34). The severity of the sign problem is then characterized by the average sign $\mathcal{S} = \overline{\langle s \rangle}$, where

$$s = \frac{W(s,x)}{\overline{W}(s,x)}, \tag{100}$$

and we have adopted the notation

$$\overline{\langle O \rangle} = \frac{\sum_s \int \mathcal{D}x \, \overline{W}(s,x) \, O}{\sum_s \int \mathcal{D}x \, \overline{W}(s,x)}. \tag{101}$$

In the absence of a sign problem $\mathcal{S} = 1$. The sign problem then becomes progressively worse as $\mathcal{S}$ approaches zero. The origin and behavior of the sign problem is not entirely understood, but it is known to be particularly sensitive to increasing the system size, inverse temperature and Hubbard interaction strength [110].

In this scenario, the reweighting method is used to extract the correct expectation value for an given observable according to

$$\langle O \rangle = \frac{\overline{\langle Os \rangle}}{\overline{\langle s \rangle}}. \tag{102}$$

At this point the Jackknife algorithm is used to correctly propagate errors. The Jackknife algorithm is a method for evaluating functions of expectation values, as in Eq. (102), and is used in conjunction with the binning method. For more information we refer the reader to Ref. [12] for a thorough description and derivation.

### 3.10  Chemical Potential Tuning

The DQMC method is formulated in the grand canonical ensemble, where the average charge density $\langle n \rangle$ is determined by the chemical potential $\mu$. The `SmoQyDQMC.jl` package provides two modes of operation to control the average particle number. The first mode is the traditional approach, where the chemical potential $\mu$ is fixed during the simulation, and $\langle n \rangle$ converges to its equilibrium value. Using this approach, one typically performs several runs at different values of $\mu$ to determine the value needed to produce the desired charge density $\langle n \rangle$. The second mode automates this process, dynamically adjusting the chemical potential $\mu$ to obtain a target value for the charge density $\langle n \rangle$ specified by the user. This automation is achieved using the algorithm described in Ref. [137] and has been implemented as a stand-alone package `MuTuner.jl` [138] that `SmoQyDQMC.jl` uses to incorporate this functionality.

The $\mu$ tuning algorithm can be utilized in simulations with and without a Fermion sign problem; however, we have found that the chemical potential tuning can become unstable if the average value of the Fermion sign becomes too small. When this occurs, we recommend reverting to the fixed $\mu$ mode of operation.

## 4  Performance

Figure 1 assesses the performance of the `SmoQyDQMC.jl` package for three representative models, namely the single-band Hubbard, Holstein, and optical SSH models, defined on 1D chains of length $\mathcal{N}$. The dimension of the system is unimportant with respect to measuring the scaling of the DQMC algorithm, which nominally scales as the cube of the total number of orbitals in the system $\mathcal{N}^3$, independent of the dimension.

Figure 1 reports the simulation run time, including the time needed to perform measurements of the time-displaced Green's function and normalized by the number of updates that were performed. All simulations were performed at half-filling ($\mu = 0$) with fixed $\Delta\tau = 0.1$ and adopting only nearest neighbor hopping $t$. For the Hubbard model, we set $U = 4t$ to place the system in the Mott insulating regime in one dimension. For the Holstein and optical SSH simulations, we set the phonon energy $\Omega = t$ and $e$-ph coupling $\alpha = t$, giving rise to charge ordered states. We note, however, that we have obtained similar performance measures when simulating low-energy optical and acoustic phonon modes [64], which are traditionally very challenging for conventional QMC approaches [123,124]. Finally, the asymmetric form for the propagator matrices was used in the Holstein and Hubbard simulations without the checkerboard approximation. For the simulations of the optical SSH model, we adopted the symmetric propagator definition and checkerboard approximation.

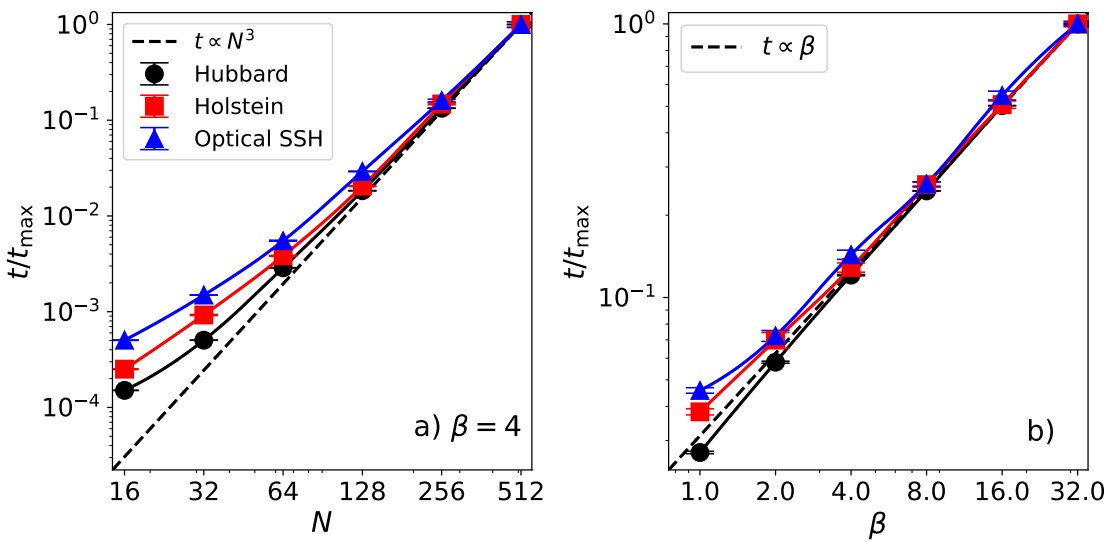

Figure 1: The average time per Monte Carlo update sweep, including the time needed for measurements of the time-displaced Green's function, for the one-dimension (1D) Hubbard (black ○), Holstein (red □), and optical SSH (blue △) chains. The left panel plots results as a function of the chain length $\mathcal{N}$ at a fixed $\beta = 4/t$. The right panel shows results as a function of inverse temperature for a fixed chain length $\mathcal{N} = 256$. All results have been normalized to the largest time and the dashed line shows the ideal $O(\beta\mathcal{N}^3)$ curve.

Figure 1 demonstrates that `SmoQyDQMC.jl` achieves the nominal $O(\beta\mathcal{N}^3)$ scaling of the DQMC algorithm for all three cases. This result confirms that our HMC updates for the phonon fields are efficient and bypass the typical increase in computational complexity normally associated with performing global or block updates of the phonon fields [67].

It is also straightforward to parallelize `SmoQyDQMC.jl` simulations with MPI using the MPI.jl package [139]. `SmoQyDQMC.jl` does not list this package as a dependency, but rather the parallelization is introduced at the script level. Examples demonstrating this functionality are included in the online documentation. Additionally, when simulations are parallelized using MPI.jl, final estimates for measured observables are obtained by averaging results over all walkers simulated in parallel.

## 5 Summary & Future Directions

The `SmoQyDQMC.jl` package, implemented in the Julia programming language, exports a user-friendly implementation of the DQMC method for simulating Hubbard and $e$-ph interactions without sacrificing performance. By adopting a scripting interface, `SmoQyDQMC.jl` is unique relative to similar DQMC software packages, opening the door to integrating the exported functionality into more complicated workflows that leverage the growing Julia ecosystem of scientific computing and machine learning packages. With extensive online documentation that includes an ever-growing list of examples, `SmoQyDQMC.jl` will help make the DQMC method accessible to a broader community of researchers.

Moving forward, one obvious direction for future development is expanding the class of Hamiltonians that `SmoQyDQMC.jl` can simulate. Adding support for inter-orbital Hubbard interactions, or user-defined Fermion interactions, and couplings to classical degrees are all

planned for future releases. Adding support for an arbitrary number of Fermion spin species, and asymmetric couplings to each spin sector is also planned. Longer term goals include developing a suite of companion packages that export other QMC variants, including the dynamical cluster approximation that uses DQMC as a solver [24,140], linear-scaling QMC methods for simulating $e$-ph models [126], the zero-temperature projector QMC method [6,113], and constrained path QMC algorithms for tackling the sign problem [30,141].

## Acknowledgements

We thank C. Miles and G. Batrouni for useful discussions and collaborations with the various algorithms utilized by this package.

**Funding information:**   This work was supported by the U.S. Department of Energy, Office of Science, Office of Basic Energy Sciences, under Award Number DE-SC0022311.

## A   Numerical Stabilization Routines

This section summarizes the numerical stabilization routines required to evaluate the various Green's function matrices in a DQMC simulation. Note that any intermediate matrix inversions are performed with an $LU$ factorization with partial pivoting.

### A.1   Routine for Stable Left Matrix Multiply

The routine outlined below updates an $LDR$ factorization from $F = LDR$ to $F' = L'D'R'$ when left multiplied by a matrix $U$:

$$F' = UF = U[\overbrace{LD}^{L_0 D_0 R_0}R] = \overbrace{L_0}^{L'}\; \overbrace{D_0}^{D'}\; \overbrace{R_0 R}^{R'} = L'D'R'. \tag{103}$$

### A.2   Routine for Stable Right Matrix Multiply

The routine outlined below updates an $LDR$ factorization from $F = LDR$ to $F' = L'D'R'$ when right multiplied by a matrix $U$:

$$F' = FU = [L\overbrace{DR]U}^{L_0 D_0 R_0} = \overbrace{LL_0}^{L'}\; \overbrace{D_0}^{D'}\; \overbrace{R_0}^{R'} = L'D'R'. \tag{104}$$

### A.3   Routine for Stable Evaluation of Eq. (38)

Below is a numerically stable routine for evaluating Eq. (38) to calculate the matrix $G_\sigma(\tau, \tau)$, where the matrices $B_\sigma(\tau, 0)$ and $B_\sigma(\beta, \tau)$ are represented by the $LDR$ factorization $F_1 = L_1 D_1 R_1$ and $F_0 = L_0 D_0 R_0$ respectively:

$$
\begin{aligned}
G =& [I + F_1 F_0]^{-1} \\
=& R_0^{-1} D_{0,\max}^{-1} [\overbrace{D_{1,\max}^{-1} L_1^\dagger R_0^{-1} D_{0,\max}^{-1} + D_{1,\min} R_1\; L_0 D_{0,\min}}^{M}]^{-1} D_{1,\max}^{-1} L_1^\dagger \\
=& R_0^{-1} D_{0,\max}^{-1} M^{-1} D_{1,\max}^{-1} L_1^\dagger.
\end{aligned}
\tag{105}
$$

## A.4 Routine for Stable Evaluation of Eq. (40)

Below is a numerically stable routine for evaluating Eq. (40) to calculate the matrix $G_\sigma(0,0)$, where the matrix $B_\sigma(\beta,0)$ is represented by the *LDR* factorization $F$:

$$
\begin{aligned}
G &= [I + F]^{-1} \\
&= R^{-1}D_{\max}^{-1}[\overbrace{R^{-1}D_{\max}^{-1} + LD_{\min}}^{M}]^{-1} \\
&= R^{-1}D_{\max}^{-1}M^{-1}.
\end{aligned}
\tag{106}
$$

## A.5 Routine for Stable Evaluation of Eq. (44b) and Eq. (45b)

Below is a numerically stable routine for evaluating Eq. (44b) to calculate the matrix $G_\sigma(\tau,0)$, where the matrices $B_\sigma(\tau,0)$ and $B_\sigma(\beta,\tau)$ are represented by the *LDR* factorization $F_1 = L_1 D_1 R_1$ and $F_0 = L_0 D_0 R_0$ respectively:

$$
\begin{aligned}
G &= [F_1^{-1} + F_0]^{-1} \\
&= R_0^{-1}D_{0,\max}^{-1}[\overbrace{D_{1,\max}^{-1}L_1^\dagger R_0^{-1}D_{1,\max}^{-1} + D_{1,\min}R_1 L_0 D_{0,\min}}^{M}]^{-1}D_{1,\min}R_1 \\
&= R_0^{-1}D_{0,\max}^{-1}M^{-1}D_{1,\min}R_1.
\end{aligned}
\tag{107}
$$

This same routine can be used to help evaluate Eq. (45b) to calculate the matrix $G_\sigma(0,\tau)$, except in this case $B_\sigma(\tau,0)$ corresponds to the factorization $F_0$ and $B_\sigma(\beta,\tau)$ corresponds to $F_1$.

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
