# Peer review of "SmoQyDQMC.jl: A flexible implementation of determinant quantum Monte Carlo for Hubbard and electron-phonon interactions"

_SciPost Physics Codebases, doi:SciPost Phys. Codebases 29-r0.3 (2024) , SciPost Phys. Codebases 29 (2024)_

## Round 2 · Referee Report · Claudio Attaccalite (Referee 1) · 2024-4-10

Strengths

Good presentation of the SmoQyDQMC.jl and its capabilities

Weaknesses

Missing a small discussion on parallelization

Report

The authors describe in detail a new determinant quantum Monte Carlo code in Julia for tightly bound Hamiltonians including on-site Hubbard and e-ph interactions. I found the manuscript to be rigorous and clear. I have only a few minor questions:

1) The approximation introduced in Eq.15 at the end is never used because it is equivalent to Eq.~14. Does it give the same result as Eq.~16? Nevertheless the authors mention it again in Eq.20/21. Which is implemented in the code? both?

2) On page 3, please explain the meaning of acronimous HMC.

3) Regarding Eq. 5, I wonder if some of the terms, the quadratic ones, in U_{disp} could be included in Eq.~3 by a redefinition of the QHO. And maybe a stupid question, where are no cubic terms in the Hamiltonian?

4) Does the factorisation of Eq. 53 mean that the matrix is written as a product of two matrices with only large/small elements? Does this help because of machine precision?

5) Can the authors write a small part about parallelization? how the code is parallelized: MPI/openMP/GPU?

6) Is there a way to slightly relax the sign condition of Eq. 34 (e.g. Phys. Rev. Lett. 80, 4558 or similar)? Is there a special case where the simulation can be done without the sign problem?

7) The footnote on page 3 is incorrect, pyALF is not a Python implementation of ALF, but only a Python interface to ALF. Please also fix ref.90 and put the package name in front of the link.

8) Did the author ever consider to import parameters from DFT codes? from example tight-binding from Wannier (https://www.wanniertools.org/theory/tight-binding-model/), Hubbard U(https://arxiv.org/abs/2203.15684), or el-ph Hamiltonian(https://arxiv.org/abs/2002.02045)? or is it not a good idea due to the approximation in DFT etc.?

Recommendation

Publish (easily meets expectations and criteria for this Journal; among top 50%)

  • validity: high
  • significance: high
  • originality: high
  • clarity: high
  • formatting: excellent
  • grammar: excellent

Author:  Steven Johnston  on 2024-04-18  [id 4433]

(in reply to Report 2 by Claudio Attaccalite on 2024-04-10)

The referee wrote: The authors describe in detail a new determinant quantum Monte Carlo code in Julia for tightly bound Hamiltonians, including on-site Hubbard and e-ph interactions. I found the manuscript to be rigorous and clear. I have only a few minor questions:

Our response: We thank the reviewer for their time and constructive comments. Below, we will provide detailed responses to each comment and outline our changes to the manuscript.

The referee wrote: 1) The approximation introduced in Eq.15 at the end is never used because it is equivalent to Eq.~14. Does it give the same result as Eq.~16? Nevertheless, the authors mention it again in Eq.20/21. Which is implemented in the code? both?

Our response: We implement both methods in SmoQyDQMC.jl. The reasons for this have to do with ensuring the efficacy of the checkerboard approximation later on in the text. Both methods result in an O(dtau^2) error. Absent the checkerboard approximation, Eq. (16) is more computationally efficient. However, using the symmetric definition introduced in Eq. (15) ensures that the propagator matrices remain Hermitian, improving the checkerboard approximation's accuracy. We have added text clarifying that both approximations are available in the code immediately following Eq. (16).

The referee wrote: 2) On page 3, please explain the meaning of acronimous HMC.

Our response: Thank you for pointing out this omission. We have corrected the text so that the meaning of the acronym HMC (hyrbid Monte Carlo) is provided.

The referee wrote: 3) Regarding Eq. 5, I wonder if some of the terms, the quadratic ones, in U_{disp} could be included in Eq.~3 by a redefinition of the QHO. And maybe a stupid question, why are no cubic terms in the Hamiltonian?

Our response: Indeed, Eq. (5) introduces additional harmonic modes that could be merged with those described by Eq. (3), and the partitioning of these terms is not unique. Characterizing the harmonic modes in the lattices would require diagonalizing the quadratic terms in the bare lattice Hamiltonian. However, this step is not helpful in practice because the code doesn't perform simulations in the lattice's eigenbasis. Instead, Eq. (3) is meant to describe only the placement of local dispersionless harmonic modes in the lattice.

There are no cubic terms on the bare lattice potential; such terms are typically forbidden by symmetry and can result in severe lattice instabilities.

The referee wrote: 4) Does the factorisation of Eq. 53 mean that the matrix is written as a product of two matrices with only large/small elements? Does this help because of machine precision?

Our response: Correct. The factorization separates the large and small scales in the matrices, further improving numerical stability. At this point, I have added a citation to the text, referring the reader to the paper that initially introduced this technique.

The referee wrote: 5) Can the authors write a small part about parallelization? how the code is parallelized: MPI/openMP/GPU?

Our response: We thank this author for this valuable suggestion. SmoQyDQMC.jl does not depend on MPI; however, it is straightforward to parallelize simulations using the MPI.jl package, and we have provided examples of how to do this in the online documentation. In these examples, analyzing the results to generate final estimates for observables will automatically average over results generated by parallel simulations performed with MPI. To highlight this functionality, we have added text discussing this at the end of the Performance section.

The referee wrote: 6) Is there a way to slightly relax the sign condition of Eq. 34 (e.g. Phys. Rev. Lett. 80, 4558 or similar)? Is there a special case where the simulation can be done without the sign problem?

Our response: There are many models that can be simulated without a sign problem. For instance, any model with only el-ph interactions and no Hubbard interaction will not have a sign problem. Any particle-hole symmetric Hamiltonian with density-density Hubbard interactions is also sign-problem-free. However, the measurements and analysis of the results are performed using the reweighting procedure, assuming there is a sign problem because this method reduces to the correct unweighted procedure if there is no sign problem.

Unfortunately, there is no straightforward way to introduce a method like the one described in Phys. Rev. Lett. 80, 4558. However, in the long term, we would like to implement a variant of the process described in Phys. Rev. B 99, 045108. But that represents a very significant development effort and is far beyond the current scope of this initial release; it would likely require developing distinct extension packages on top of or adjacent to SmoQyDQMC.jl.

The referee wrote: 7) The footnote on page 3 is incorrect, pyALF is not a Python implementation of ALF, but only a Python interface to ALF. Please also fix ref.90 and put the package name in front of the link.

Our response: Thank you for pointing out this error. We have corrected it in the text.

The referee wrote: 8) Did the author ever consider to import parameters from DFT codes? from example tight-binding from Wannier (https://www.wanniertools.org/theory/tight-binding-model/), Hubbard U(https://arxiv.org/abs/2203.15684), or el-ph Hamiltonian(https://arxiv.org/abs/2002.02045)? or is it not a good idea due to the approximation in DFT etc.?

Our response: This functionality is a longer-term aspiration goal. We envision the flexibility of SmoQyDQMC.jl allowing for the simulation of more physically realistic Hamiltonians and plan to add functionality in future releases that work toward this goal. Including ways to interface with DFT calculations and define Hamiltonians based on DFT results is an important aspect of this effort. With respect to the Julia ecosystem specifically, we are particularly interested in seeing if there is a way to interface SmoQyDQMC.jl with the DFTK.jl package that is being actively developed (https://github.com/JuliaMolSim/DFTK.jl.git). However, explicit functionality to do this is beyond the scope of this initial release.

---

## Round 2 · Referee Report · Anonymous (Referee 2) · 2024-4-11

Strengths

1- The code is flexible and modular, allowing for seamless integration into one's workflow through the scripting interface of the Julia implementation. 2- The authors provide supporting packages that implement the DQMC kernel and various observables. 3- The package supports a broad range of phonon and electron-phonon Hamiltonians.

Weaknesses

1- The description of the DQMC algorithm could be more comprehensive. 2- A general algorithm outlining the structure of the package is not provided. 3- No specific examples are given to illustrate the usage of the package.

Report

The article presents a package developed in Julia for executing determinant quantum Monte Carlo (DQMC) simulations on model Hamiltonians. These Hamiltonians are defined on a lattice and encompass Hubbard, phonon, and electron-phonon interactions. The article comes with comprehensive documentation, and the source code is well-structured and annotated.

The package's scripting interface facilitates seamless parallelization and integration into a Julia workflow, a language that is gaining traction in the scientific programming community. Additionally, the authors offer two supplementary packages: one for the DQMC kernel and another for various observables. These packages enable users to create their own DQMC algorithm implementations.

However, the primary critique is that the article is challenging to comprehend due to the brief explanation of the DQMC algorithm in section 3.1. It's unclear how the specific algorithms outlined in sections 3.3 through 3.7 fit within the broader DQMC algorithm. Therefore, I recommend expanding section 3.1 to include a detailed algorithmic description of the DQMC process.

Another missing element is an illustrative example. To validate the specific algorithms discussed in sections 3.3 through 3.7, I suggest incorporating relevant examples. Below, I provide a list of additional, more specific changes that could be made to improve the article.

Requested changes

1- It is not clear whether the JDQMCFrameworks and JDQMCMeasurements are also integrated in the SmoQyDQMC package 2- $\hat{\nu}_{ph}$ is not defined after Eq. 11 3- The meaning of $n_{i,a}$ is not clear to me in Eq. 31 4- The importance of singe-particle Green's functions defined in section 3.2 becomes clear only in later sections, maybe the authors could comment on the use of Green's functions in DQMC already in section 3.2 5- Should it be $B_{\sigma,l'+1}$ In Eq. 39? 6-The authors could illustrate Eq. 57 with an example for clarity. 7-Should it be $k_{\sigma,b}$ before Eq. 60? 8-The authors should pay attention to repeated the use of symbols. For example, $M$ is used for both: mass and fermion matrix.

  • validity: high
  • significance: good
  • originality: good
  • clarity: ok
  • formatting: excellent
  • grammar: excellent

Author:  Steven Johnston  on 2024-04-18  [id 4432]

(in reply to Report 1 on 2024-04-11)
Category:
answer to question

The referee wrote:
The article presents a package developed in Julia for executing determinant quantum Monte Carlo (DQMC) simulations on model Hamiltonians. These Hamiltonians are defined on a lattice and encompass Hubbard, phonon, and electron-phonon interactions. The article comes with comprehensive documentation, and the source code is well-structured and annotated.

The package's scripting interface facilitates seamless parallelization and integration into a Julia workflow, a language that is gaining traction in the scientific programming community. Additionally, the authors offer two supplementary packages: one for the DQMC kernel and another for various observables. These packages enable users to create their own DQMC algorithm implementations.

However, the primary critique is that the article is challenging to comprehend due to the brief explanation of the DQMC algorithm in section 3.1. It's unclear how the specific algorithms outlined in sections 3.3 through 3.7 fit within the broader DQMC algorithm. Therefore, I recommend expanding section 3.1 to include a detailed algorithmic description of the DQMC process.

Another missing element is an illustrative example. To validate the specific algorithms discussed in sections 3.3 through 3.7, I suggest incorporating relevant examples. Below, I provide a list of additional, more specific changes that could be made to improve the article.

Our response:
We thank the reviewer for their time and constructive comments. Below, we will provide detailed responses to each of their comments and outline our changes to the manuscript.

The referee wrote:
However, the primary critique is that the article is challenging to comprehend due to the brief explanation of the DQMC algorithm in section 3.1. It's unclear how the specific algorithms outlined in sections 3.3 through 3.7 fit within the broader DQMC algorithm. Therefore, I recommend expanding section 3.1 to include a detailed algorithmic description of the DQMC process.

Our response:
Thank you for this helpful critique, which we fully agree with. We have added a new section entitled "DQMC Simulation Overview" following section 3.1. This new section outlines the structure of a DQMC simulation, referencing the later subsections where necessary. Including this new subsection should help contextualize the discussion of the later algorithms for people reading the paper.

The referee wrote:
Another missing element is an illustrative example. To validate the specific algorithms discussed in sections 3.3 through 3.7, I suggest incorporating relevant examples.

Our response:
We intend for readers to refer to the online documentation for worked examples, as the online documentation already contains an exensive list of examples. These examples are also sorted in an order that approximately increases with complexity, with the very first example being a simple 1D Hubbard chain at half-filling. We have chosen this structure because we want the documentation to be a living document, and we will be adding new examples as we conduct new research with this code. In the new "DQMC Simulation Overview" section, we encourage readers to refer to the online documentation for examples.

The referee wrote:
It is not clear whether the JDQMCFrameworks and JDQMCMeasurements are also integrated in the SmoQyDQMC package

Our response:
The JDQMCFrameworks.jl and JDQMCMeasurements.jl packages are integrated into the SmoQyDQMC.jl package. More succinctly, the SmoQyDQMC.jl package was implemented using the functionality exported by these other packages. We now state this explicitly in the last sentence of section 1.2.

The referee wrote:
\hat{\nu}_{\rm ph} is not defined after Eq. 11.

Our response:
We thank the author for catching this typo; we have corrected it.

The referee wrote:
The meaning of n_{i,a} is not clear to me in Eq. 31.

Our response:
Once again, we thank the author for catching this typo. The \alpha in equation (31) has been corrected to read \nu.

The referee wrote:
The importance of singe-particle Green's functions defined in section 3.2 becomes clear only in later sections, maybe the authors could comment on the use of Green's functions in DQMC already in section 3.2.

Our response:
In the new section "DQMC Simulation Overview," we specifically reference the single-particle electron Green's function matrix, mentioning that it is essential for measuring the electronic correlation function. We also refer readers to (now) section 3.3 to see how it is defined.

The referee wrote:
Should it be B_{\sigma,l^\prime+1} in Eq. 39?

Our response:
Yes, that is correct. This definition allows for the correct limiting definition $B_\sigma(\tau,\tau-\Delta\tau) = B_{\sigma,l}$ and $B_\sigma(\tau,\tau) = I$.

The referee wrote:
The authors could illustrate Eq. 57 with an example for clarity.

Our response:
Our intention is for readers to refer to the online documentation for examples. When they run any of the example simulations, the maximum numerical error $\delta G$ that occurs during the simulation is reported.

The referee wrote:
Should it be k_{\sigma, b} before Eq. 60?

Our response:
Yes, we thank the author for pointing this out. We have corrected the equation in the updated manuscript.

The referee wrote:
The authors should pay attention to repeated the use of symbols. For example, M, is used for both: mass and fermion matrix.

Our response:
We thank the referee for raising this issue. It is unfortunate but true that most of the DQMC community uses $M$ to denote the Fermion matrix, while most model Hamiltonian communities also denote ion masses with a $M$. We prefer to keep our notation in line with this convention. The differences in these quantities should be apparent in our discussion from the context and subscripts attached to each. For example, our fermion matrices always have a spin index for subscripts, while the ion masses always have mode indicates attached to them. Having said that, we found one place where the context may need to be clarified (in the current Sec. 3.7.2), and we have added a sentence reminding the reader of the notation.

---

## Round 3 · Referee Report · Anonymous (Referee 2) · 2024-4-24

Report

All the questions and comments have been addressed by the authors, and the quality of the manuscript has been significantly improved. Therefore, I highly recommend the manuscript for publication in the SciPost journal.

Recommendation

Publish (easily meets expectations and criteria for this Journal; among top 50%)

  • validity: high
  • significance: high
  • originality: high
  • clarity: high
  • formatting: excellent
  • grammar: excellent

Author:  Steven Johnston  on 2024-04-24  [id 4443]

(in reply to Report 1 on 2024-04-24)

We again thank the referee for their time and for their positive recommendation.

---

## Round 3 · Author Response

We would like to thank the referees for their time spent reviewing our work and for their constructive comments. We have made several changes to the manuscript in response as detailed below and in our responses to the referees.

---

## Round 3 · List of Changes

1) We have corrected the typos pointed out by the referees.
2) We have added a new section providing an overview of a typical DQMC simulation.
3) We have added some sentences clarifying the notation in several places.
4) We have added some discussion on MPI parallelization.
5) We have added several links to relevant online documentation.

---

## Editorial Decision

published